# Resting mitochondrial complex I from *Drosophila melanogaster* adopts a helix-locked state

**Abhilash Padavannil[1], Anjaneyulu Murari[2], Shauna-Kay Rhooms[2], Edward Owusu-Ansah[2,3], James A Letts[1]***

[1]Department of Molecular and Cellular Biology, University of California, Davis, Davis, United States; [2]Department of Physiology and Cellular Biophysics, Columbia University Irving Medical Center, New York, United States; [3]The Robert N. Butler Columbia Aging Center, Columbia University Irving Medical Center, New York, United States

**\*For correspondence:**
jaletts@ucdavis.edu

**Competing interest:** The authors declare that no competing interests exist.

**Abstract** Respiratory complex I is a proton-pumping oxidoreductase key to bioenergetic metabolism. Biochemical studies have found a divide in the behavior of complex I in metazoans that aligns with the evolutionary split between Protostomia and Deuterostomia. Complex I from Deuterostomia including mammals can adopt a biochemically defined off-pathway 'deactive' state, whereas complex I from Protostomia cannot. The presence of off-pathway states complicates the interpretation of structural results and has led to considerable mechanistic debate. Here, we report the structure of mitochondrial complex I from the thoracic muscles of the model protostome *Drosophila melanogaster*. We show that although *D. melanogaster* complex I (*Dm*-CI) does not have a NEM-sensitive deactive state, it does show slow activation kinetics indicative of an off-pathway resting state. The resting-state structure of *Dm*-CI from the thoracic muscle reveals multiple conformations. We identify a helix-locked state in which an N-terminal α-helix on the NDUFS4 subunit wedges between the peripheral and membrane arms. Comparison of the *Dm*-CI structure and conformational states to those observed in bacteria, yeast, and mammals provides insight into the roles of subunits across organisms, explains why the *Dm*-CI off-pathway resting state is NEM insensitive, and raises questions regarding current mechanistic models of complex I turnover.

## Editor's evaluation

This important study offers new insights into the structure and function of respiratory complex I. Based on convincing cryoEM data for the enzyme complex from the insect model organism *Drosophila melanogaster*, the authors discuss the functional significance of two major conformational states. This study is relevant to readers interested in the evolution and molecular mechanisms of respiratory chain complexes, as well as mitochondrial diseases and the development of new insecticides.

## Introduction

The final stage of eukaryotic cellular respiration occurs in mitochondria. The terminal respiratory reactions are catalyzed by the oxidative phosphorylation (OXPHOS) electron transport chain (ETC) – a series of large membrane protein complexes that reside in the inner mitochondrial membrane (IMM). Several ETC complexes are redox-coupled $H^+$ pumps that connect oxygen consumption to adenosine triphosphate (ATP) synthesis by the generation of a proton motive force (pmf) across the IMM that

is used to power the ATP synthase complex. The generation of the pmf is driven by the transfer of electrons from reduced substrates (NADH and succinate) to $O_2$ via four ETC complexes (Complexes I-IV, CI-IV) and electron carriers ubiquinone (coenzyme Q, CoQ) and cytochrome *c* (cyt *c*). In addition to their independent existence, ETC complexes can form higher-order structures known as super-complexes (SCs) (*Letts and Sazanov, 2017*; *Schägger and Pfeiffer, 2000*). Across species, the most common SCs are formed between CI, a dimer of CIII ($CIII_2$) and CIV (SC $I+III_2+IV$) *Letts et al., 2016b*; CI and $CIII_2$ alone (SC $I+III_2$) *Letts et al., 2019*; and $CIII_2$ and CIV (SC $III_2 +IV$) (*Hartley et al., 2018*; *Rathore et al., 2018*; *Vercellino and Sazanov, 2021*). The advantage of SC formation remains unde-fined, but they are found across eukaryotes, often as the most abundant form of the ETC complexes (*Davies et al., 2018*; *Schägger and Pfeiffer, 2001*; *Zhou et al., 2022*).

CI couples the transfer of electrons from NADH to CoQ to the pumping of four $H^+$ across the IMM (*Galkin et al., 1999*; *Jones et al., 2017*). It has an 'L' shaped structure consisting of two arms: a peripheral arm (PA) that extends into the matrix and a membrane arm (MA) that is embedded in the IMM. CI accepts electrons from NADH onto an FMN co-factor near the distal tip of the PA and transfers them to CoQ via a series of seven iron-sulfur (FeS) clusters. With few exceptions, the catalytic core of 14 subunits is conserved across species and contains all the redox cofactors and active sites required for catalysis (*Hirst, 2013*; *Sazanov, 2015*). In addition to the core subunits, eukaryotic CI has varying numbers of accessory subunits, e.g., 29 in the yeast *Yarrowia lipolytica* and 31 in mammals (*Carroll et al., 2003*; *Letts and Sazanov, 2015*; *Padavannil et al., 2021*; *Parey et al., 2019*). The accessory subunits are needed for the assembly and stability of the complex (*Garcia et al., 2017*; *Stroud et al., 2016*) and in some cases may play an active role in regulating turnover (*Padavannil et al., 2021*).

The molecular mechanism of the coupling between electron transfer and proton pumping has been the target of much research and debate (*Chung et al., 2022a*; *Kampjut and Sazanov, 2022*). None-theless, thanks to a plethora of high-resolution structures, a framework for the coupling mechanism is emerging in which specific conformational changes in the CoQ binding site loops at the interface of the PA and MA initiate a wave of conformational changes and electrostatic interactions upon CoQ reduction that propagate along the MA via a hydrophilic axis of amino acid residues resulting in $H^+$ pumping (*Kampjut and Sazanov, 2020*; *Kravchuk et al., 2022*; *Zickermann et al., 2015*). Although this model is consistent with most mutagenesis and structural data, more experiments are needed to confirm the predictions of the model and examine possible variations in coupling and regulation across organisms (*Klusch et al., 2021*; *Maldonado et al., 2020*; *Zhou et al., 2022*).

Complicating the structural elucidation of the coupling mechanism is the fact that resting CI from yeast and mammals used for detailed structural studies have been shown to exist in two distinct biochemical states: the catalytically competent active (A) state and the off-pathway deactive (D) state (*Gavrikova and Vinogradov, 1999*; *Maklashina et al., 2003*). In these species, CI spontaneously undergoes an A-to-D transition when exposed to physiological temperatures in the absence of reduced substrates (*Babot et al., 2014*). Biochemically, the D state is characterized by a solvent-exposed cysteine residue on the transmembrane helix 1–2 loop of the ND3 core subunit (TMH1-$2^{ND3}$) that can be modified by thiol-reactive agents such as N-ethyl maleimide (NEM) (*Galkin et al., 2008*). Modification of the cysteine residue traps CI in the D state. The presence of the D state is thought to protect cells from ROS-mediated damage due to CI reverse electron transport (*Chouchani et al., 2013*). It remains unclear whether conformations of CI in the presence (*Kampjut and Sazanov, 2020*) or absence (*Agip et al., 2018*; *Blaza et al., 2018*) of added substrates, which differ in the structure of ND3's cysteine-containing loop and in the relative positions of the PA and MA, correspond to the A and D states of CI or if they are part of CI's catalytic cycle (*Chung et al., 2022a*; *Kampjut and Sazanov, 2022*).

Nonetheless, the A-to-D transition is not universally conserved across species (*Maklashina et al., 2003*). Biochemical characterization of CI from bacteria and more recently the ciliate *Tetrahymena thermophila* has failed to detect a deactive state using the standard NEM approach (*Maklashina et al., 2003*; *Zhou et al., 2022*). Structural analysis of the *T. thermophila* CI demonstrated that its PA/MA interface is more extensive than in other species (*Zhou et al., 2022*), likely precluding the opening of the complex in the same manner observed in mammals (*Agip et al., 2018*; *Kampjut and Sazanov, 2020*) and *E. coli* (*Kolata and Efremov, 2021*; *Kravchuk et al., 2022*). Although fungi and metazoans diverged approximately 1300 million years ago they form a broad clade of eukaryotes

known as the opisthokonts and are thus more closely related to each other than either is to the ciliate *T. thermophila*. This suggests that the A-to-D transition may be a biochemical feature of CI that evolved in early opisthokonts. However, even within the opisthokonts the A-to-D transition is not universally detected. An NEM-sensitive D-state is seen in the CI of all characterized fungi but within metazoans, it is observed in deuterostomes but absent in protostomes (*Figure 1A*; *Maklashina et al., 2003*). Deuterostomes and protostomes diverged after the evolution of bilaterians approximately 600 million years ago (*Dunn et al., 2014*). This indicates either the loss of an ancestral NEM-sensitive D-state in Protostomia or the evolution of distinct, though biochemically similar, D-states in Fungi and Deuterostomia. It is important to note that across opisthokonts, and eukaryotes more broadly, CI from very few species have been biochemically characterized in detail. Thus, the evolution of this biochemical feature of the complex remains poorly understood.

Further structural and biochemical analyses of CI in organisms without an A-to-D transition are needed to determine whether the observed conformational changes are catalytic or part of the off-pathway deactive state. Specifically, structural and functional characterization of a protostomian CI, like that from the model insect *D. melanogaster*, would be an important target due to its relatively close evolutionary relationship to the most well-characterized mammalian complexes. *D. melanogaster* is a genetically tractable system, has a similar ETC to humans, and is an emerging model for CI assembly and regulation (*Garcia et al., 2017*; *Murari et al., 2022*; *Murari et al., 2021*; *Murari et al., 2020*; *Rhooms et al., 2019*; *Xu et al., 2019*). Furthermore, CI is an established target for insecticides and agricultural pesticides (*Murai and Miyoshi, 2016*). Particularly with the emergence of *Drosophila suzukii* as a major pest for soft summer fruits (*Tait et al., 2021*), it is important to elucidate the structure of CI from a *Drosophila* species to identify unique aspects of CI in this genus. Here, we report the functional and structural characterization of thoracic muscle *Dm*-CI.

## Results

### *D. melanogaster* CI possesses a NEM-insensitive off-pathway resting state

We assayed for the presence of an A-to-D transition in *Dm*-CI from thoracic muscles using the established NEM sensitivity assay on isolated mitochondrial membranes (*Figure 1B*; *Babot et al., 2014*; *Galkin et al., 2008*). In this assay, incubation of the mitochondrial membranes at 37 °C in the absence of substrate should deactivate CI resulting in increased sensitivity to NEM inhibition. The impact of NEM incubation can be reduced by re-activation of the complex with a small amount of NADH prior to the addition of the NEM. When compared to mammalian mitochondrial membranes (*Sus scrofa*, *Figure 1C*), the *D. melanogaster* steady-state NADH oxidation rate is unaffected by NEM after incubation at elevated temperatures (*Figure 1B and C*). This is consistent with the lack of an A-to-D transition in *Dm*-CI as has been seen in the other protostomians, namely *Lumbricus terrestris* (earth worm), *Homarus americanus* (lobster), and *Acheta domesticus* (cricket) (*Figure 1A*; *Maklashina et al., 2003*). Moreover, although the addition of NADH prior to NEM rescues activity in *Sus scrofa* (*Figure 1C*), pre-activation with NADH sensitizes *Dm*-CI to NEM treatment, significantly reducing the rate of NADH oxidation after the addition of 2 mM NEM (*Figure 1B*). This result is consistent with a model in which the TMH1-2$^{ND3}$ cysteine remains buried in resting *Dm*-CI but becomes accessible during turnover (*Kampjut and Sazanov, 2022*). Exposure of the ND3 cysteine during turnover has also been recently observed in *Bos taurus* CI (*Burger et al., 2022*).

When the time course of NADH oxidation is examined for deactivated porcine mitochondrial membranes an activation lag phase is observed in which the rate of NADH oxidation progressively increases until steady state turnover is achieved (*Figure 1D and F*). For the Porcine sample activation can be blocked by the addition of NEM or slowed by the addition of $MgCl_2$ (*Figure 1D and F*). In the case of deactivated *Drosophila* mitochondrial membranes an activation lag phase is also observed; however, the rate of activation is insensitive to the presence of either NEM or $MgCl_2$ (*Figure 1E and G*). Importantly, the activation lag is not observed after pre-activation with 5 µM of NADH (*Figure 1E and G*). These results are consistent with *Dm*-CI adopting an off-pathway resting state that is insensitive to both NEM and $MgCl_2$, making it biochemically distinct from the deactive state seen in Deuterostomes and Fungi.

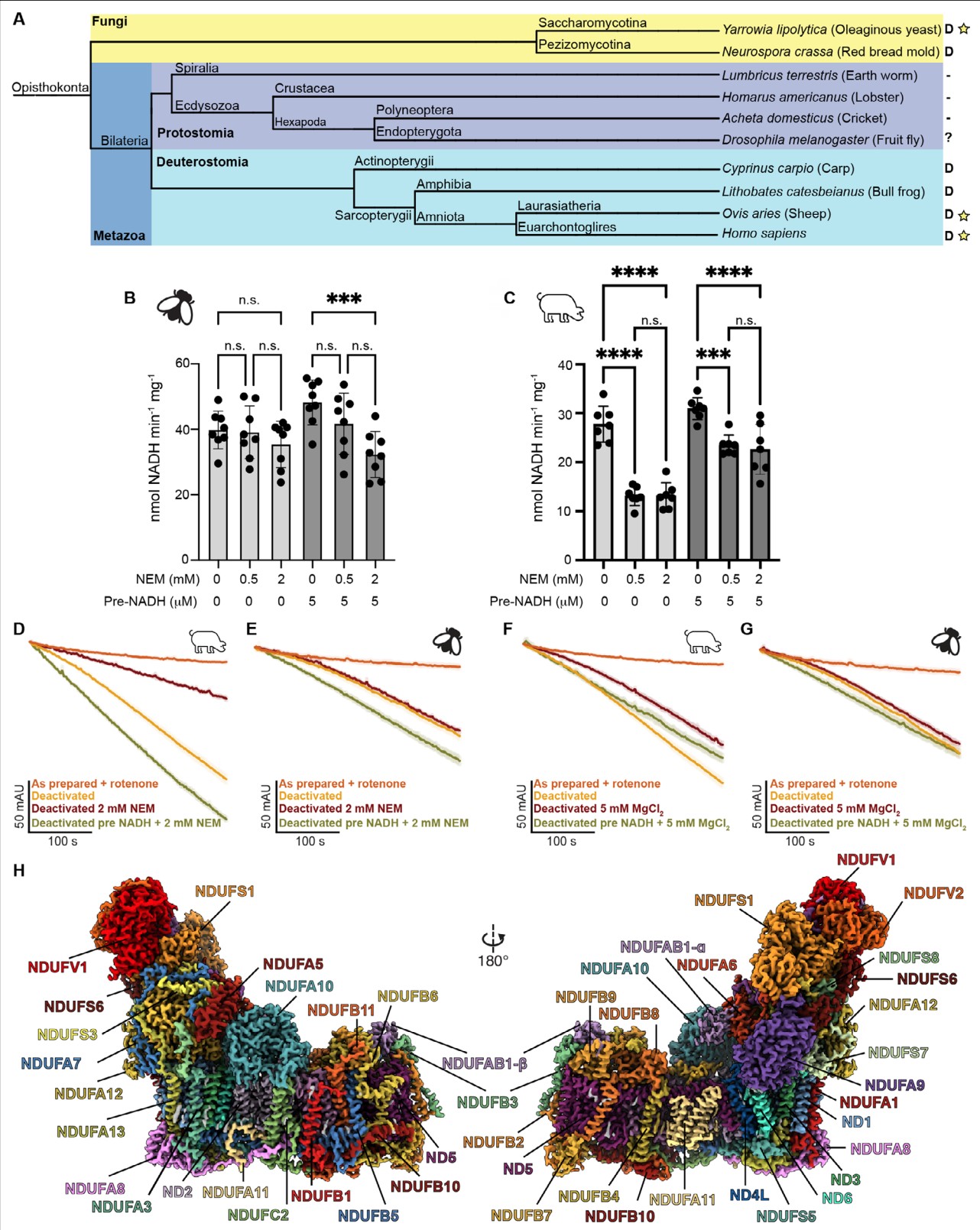

**Figure 1.** Evolution, biochemical characterization, and structure of *D. melanogaster* mitochondrial CI. (**A**) Dendrogram showing biochemically characterized CI from Opisthokonts. Distinct groups are highlighted with Fungi in yellow and metazoans in blue (Protostomia in dark blue and Deuterostomia in cyan). 'D' indicates the presence of a NEM-sensitive deactive state in the CI of the species. A minus sign indicates the absence of a NEM-sensitive deactive state in the CI of the species. A question mark indicates that the species has not been biochemically characterized for the

*Figure 1 continued on next page*

*Figure 1 continued*

presence of the deactive state. A star indicates that structures of CI from that species are currently available. (**B, C**) Functional characterization of A-to-D transition in isolated mitochondrial membranes of *D. melanogaster* (**B**) and *S. scrofa* (**C**) by spectroscopic measurement of NADH dehydrogenase activity at 340 nm in the presence of the indicated concentrations of N-ethylmaleimide (NEM) and pre-activation with 5 µM NADH or water. Individual values, average, and SEM are shown, n=7–8. Statistical analysis with ANOVA with Šídák's multiple comparisons tests. **p<0.01; ****p<0.0001; n.s. denotes not significant. (**D–G**) Time courses of NADH oxidation reveal an off-pathway resting state in *Dm*-CI. NADH oxidation was measured by the change in absorbance at 340 nm of deactivated CI in mitochondrial membranes with or without pre-activation with 5 µM NADH. (**D**) S. *scrofa* with or without 2 mM NEM (n=3–4). (**E**) *D. melanogaster* with or without 2 mM NEM (n=3–5). (**F**) *S. scrofa* with or without 5 mM MgCl₂ (n=3). (**G**) *D. melanogaster* with or without 5 mM MgCl₂ (n=3–4). The average time course is plotted for each condition with standard error for each timepoint shown as the shaded boundary. (**H**) Cryo-EM reconstruction of *Dm*-CI colored by subunit.

The online version of this article includes the following source data and figure supplement(s) for figure 1:

**Figure supplement 1.** Isolation and characterization of detergent solubilized CI from *D. melanogaster* thoracic mitochondria.

**Figure supplement 1—source data 1.** Source data for Blue-native PAGE (BN-PAGE) of fractions from (*Figure 1—figure supplement 1A*) visualized by CI in gel activity staining.

**Figure supplement 2.** Cryogenic electron microscopy (CryoEM) image processing and refinement.

**Figure supplement 3.** Overall structure of *Dm*-CI.

**Figure supplement 4.** *Dm*-CI electron transfer pathway, co-factors, and hydrophilic axis.

**Figure supplement 5.** Detection of NDUFA2 by Western blotting.

**Figure supplement 5—source data 1.** Source data for mitochondrial preparations from thoraces of 2-day-old flies with the genotypes indicated were analyzed by BN-PAGE, followed by Western blotting with the antibodies shown.

**Figure supplement 6.** NDUFA3 of *Dm*-CI.

**Figure supplement 7.** Structural comparison of CI core subunits from *Y.lipolytica* (PDB: 6RFR), *D. melanogaster* (this study), and *O. aries* (PDB:6ZKC).

To further characterize the differences in activity, *Ss*-CI and *Dm*-CI were extracted from washed mitochondrial membranes using the mild detergent digitonin, followed by exchange into glyco-diosgenin (GDN) and enrichment by sucrose gradient ultracentrifugation (*Figure 1—figure supplement 1A*). Fractions containing CI activity, as assessed by blue-native polyacrylamide gel electrophoresis (BN-PAGE) in-gel activity (*Figure 1—figure supplement 1B*), were pooled and concentrated. NADH oxidation rates for these partially purified samples were determined to be 2.87 ± 0.08 µmol NADH min⁻¹ mg⁻¹ (mean ± SEM, n=4) for *Dm*-CI and 1.77 ± 0.10 µmol NADH min⁻¹ mg⁻¹ (mean ± SEM, n=5) for *Ss*-CI prepared in the same manner. We also tested the sensitivity of the detergent-solubilized complexes to NEM and MgCl₂ after incubation at 37 °C in the absence of substrate (*Figure 1—figure supplement 1C–F*). These data were consistent with the activity in the mitochondrial membranes in that *Ss*-CI showed a clear sensitivity to both NEM and MgCl₂ and *Dm*-CI did not (*Figure 1—figure supplement 1C–F*). However, the activation lag for the *Dm*-CI was less evident in the detergent-solubilized sample suggesting that activation may be faster in detergent. To better understand the functional differences between protostomian and deuterostomian CI, we solved the structure of thoracic muscle *Dm*-CI by single-particle cryogenic electron microscopy (cryoEM).

## Overall structure of mitochondrial CI from *D. melanogaster* thoracic muscle

Samples for structure determination were prepared similarly to the sample used in activity measurements except that the digitonin-extracted complexes were exchanged into the amphipathic polymer (amphipol) A8-35 instead of GDN. The amphipol-stabilized samples were applied directly to EM grids after partial purification on a sucrose gradient and used for cryoEM data collection (*Table 1*). The structure of *Dm*-CI was resolved to a nominal resolution of 3.3 Å (*Figure 1H*, *Video 1*, *Figure 1—figure supplement 2*, *Table 1*). This partially purified sample also contained particles of *Dm*-CIII₂ and *Dm*-ATPases, albeit in insufficient numbers for high-resolution reconstruction (*Figure 1—figure supplement 2*). Consistent with previous studies on *Dm*-CI assembly (*Garcia et al., 2017*), but in contrast to what is seen in mammalian cardiac mitochondria, we did not observe large amounts of SC I+III₂ either biochemically on the BN-PAGE gels or as particles in the cryoEM dataset (*Figure 1—figure supplements 1B and 2*).

Our *Dm*-CI structure contained the 14 core subunits as well as 29 accessory subunits (*Figure 1D* and *Figure 1—figure supplement 3*), for a total composition of 43 subunits, two fewer than the 45 total

**Table 1.** Data collection and image processing.

| Data collection and image processing | | |
|---|---|---|
| Microscope | TFS Glacios | |
| Voltage(kV) | 200 | |
| Camera | K3 | |
| Data collection software | Serial EM | |
| Magnification | 56818 | |
| Electron exposure (e/Å²) | 60 | |
| Exposure time (s) | 3 | |
| Frame number | 75 | |
| Defocus range (μm) | –0.5 to –3.0 | |
| Super resolution pixel size (Å) | 0.44 | |
| Number of micrographs | 11066 | |
| EMPIAR accession code | | |
| | Helix-locked state | Flexible class 1 state |
| Number of particles for final reconstruction | 64,806 | 72,611 |
| Resolution focused refinement (Å) | 3.4 Å | 3.4 Å |
| EMDB accession code | 28582 | 28581 |
| | | |
| **Model refinement statistics** | | |
| Manual modelling software | Coot | |
| Refinement software | Phenix | |
| **Cross-correlation** | | |
| Mask | 0.84 | 0.84 |
| Volume | 0.81 | 0.80 |
| **Model composition** | | |
| Non-hydrogen atoms | 67,760 | 66,739 |
| Protein residues | 8141 | 8090 |
| Ligands | 68 | 49 |
| **Ramachandran** | | |
| Favored (%) | 97.00 | 97.0 |
| Allowed (%) | 3.00 | 3.00 |
| Outlier (%) | 0.00 | 0.00 |
| Rotamer outliers (%) | 0.9 | 0.7 |
| Clash score | 6 | 5 |
| **RMSD** | | |
| Bond length(Å) | 0.005 | 0.005 |
| Bond angle (°) | 0.973 | 0.962 |
| **B factors (Å² min/max/mean)** | | |
| Protein | 24.39/136.88/57.00 | 5.74/81.66/25.84 |
| Ligands | 26.80/91.61/59.37 | 7.72/57.56/30.70 |
| MolProbity score | 1.44 | 1.42 |
| Average atom inclusion | 0.69 | 0.68 |
| Q-score | 0.50 | 0.50 |
| PDB Accession Code | **8ESZ** | **8ESW** |

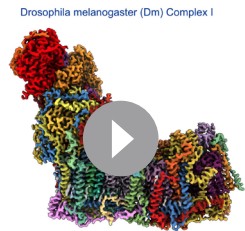

Drosophila melanogaster (Dm) Complex I

**Video 1.** Cryogenic electron microscopy (CryoEM) density map and model of *D. melanogaster* CI. The subunits are colored as in *Figure 1*.

https://elifesciences.org/articles/84415/figures#video1

subunits observed in mammals (*Padavannil et al., 2021*). Unlike *Y. lipolytica*, *T. thermophila,* and plant CI, there were no accessory subunits unique to *Dm*-CI. The two missing subunits are NDUFA2, and NDUFC1. An accessory subunit consistent with the position of NDUFV3 in mammals was present at sub-stoichiometric levels (see below). The electron transfer pathway from FMN to the final N2 cluster was conserved (*Figure 1—figure supplement 4A, B*). Although no quinone was added to the CI preparation, we were able to build a quinone molecule into density in the Q-tunnel (*Figure 1—figure supplement 4C*). The E-channel and hydrophilic axis residues that are key to the coupling of electron transfer to proton pumping were also conserved (*Figure 1—figure supplement 4A, H*; *Baradaran et al., 2013*).

The metazoan-specific transmembrane (TM) accessory subunit NDUFC1 is absent from the *Dm*-CI structure, consistent with the lack of a known ortholog (*Garcia et al., 2017*). Conversely, the N-module subunit NDUFA2, which is seen in all other known eukaryotic CI structures (*Fiedorczuk et al., 2016*; *Maldonado et al., 2020*; *Parey et al., 2019*; *Zhou et al., 2022*), is missing from the *Dm*-CI structure. A *D. melanogaster* ortholog of NDUFA2 (CG15434), which has a thioredoxin fold, has been described and is expressed in muscles (*Garcia et al., 2017*; *Li et al., 2022*; *Figure 1—figure supplement 5A, B*). However, the majority of a FLAG-tagged NDUFA2 transgene failed to incorporate into CI (*Figure 1—figure supplement 5A, B*). Evidently, future studies are required to fully elucidate the conditions under which NDUFA2 is incoporated into the complex.

An ortholog of the accessory subunit NDUFA3 was not annotated in the *D. melanogaster* proteome (*Garcia et al., 2017*). However, we identified density consistent with the presence of an NDUFA3 ortholog (*Figure 1H*, *Video 2* and *Figure 1—figure supplement 6*). Amino acid assignment based on the observed side chain density allowed for the identification of an uncharacterized protein Dme1_CG9034, isoform B as the most likely NDUFA3 ortholog in *D. melanogaster* (*Figure 1—figure supplement 6A, B*). Consistent with this observation, proteomic analysis of a *Dm*-CI band cut from blue native gels identified CG9034 as one of the proteins that co-migrates with *Dm*-CI (*Garcia et al., 2017*); however, because it was not annotated and bioinformatics searches failed to identify it as an NDUFA3 ortholog, it was not identified as such.

While there were notable differences in the *Dm*-NDUFS1, *Dm*-NDUFS7, *Dm*-ND2, and *Dm*-ND5 structures (discussed below), the *Dm*-CI core subunits were overall like the core subunits of other opisthokonts (*Figure 1—figure supplement 7*). Differences in accessory subunits NDUFA11, NDUFC2, NDUFA10, NDUFB4, and NDUFB9 suggest that there may be some differences in the mechanism of assembly and regulation of *Dm*-CI and mammalian CI, reveal how accessory subunits may influence CI in mammals and reveal why *D. melanogaster* mitochondria have less SC I+III$_2$ than mammals.

## Features of *Dm*-CI subunits with implications for assembly and stability

*NDUFA2* – Whereas the loop formed by amino acid residues 665–685 of NDUFS1 forms an α-helix in other species, in *Dm*-CI it is a poorly resolved coil lacking secondary structure (*Figure 2A*, *Video 3* and *Figure 2—figure supplement 1A*). This region forms part of the interface with accessory subunit NDUFA2 in other species, and the loss of the α-helix may be responsible for the absence of NDUFA2 in the *Dm*-CI structure (*Figure 2—figure*

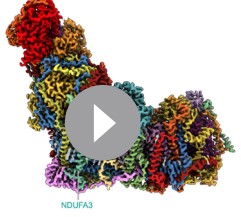

NDUFA3

**Video 2.** Cryogenic electron microscopy (CryoEM) density map of *D. melanogaster* CI. Subunit NDUFA3 identified from the map is highlighted. The subunits are colored as in *Figure 1*.

https://elifesciences.org/articles/84415/figures#video2

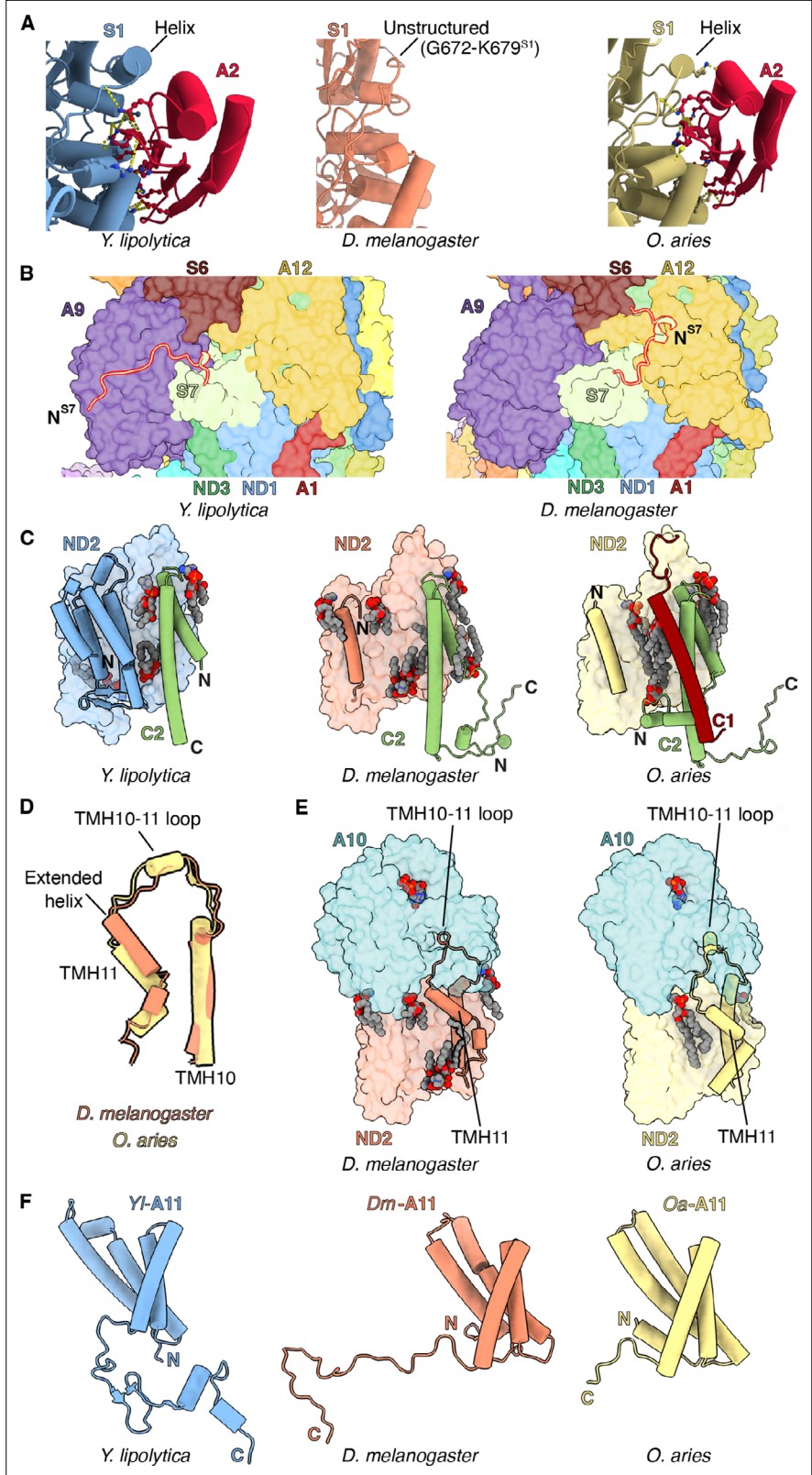

**Figure 2.** Features of *Dm*-CI subunits that may impact assembly and stability. (**A**) NDUFS1-NDUFA2 interface in *Y. lipolytic* (PDB: 6YJ4), *D. melanogaster* (this study), and *O. aries* (PDB:6ZKC) are shown. (**B**) The N-terminal extension of NDUFS7 in *Y. lipolytica* (PDB:6YJ4) and *D. melanogaster* are shown as cartoons. The other subunits are shown in surface colored as in *Figure 1*. (**C**) *Y. lipolytic* (PDB: 6YJ4), *D. melanogaster* (this study), and *O. aries* (PDB:6ZKC)

*Figure 2 continued on next page*

*Figure 2 continued*

ND2 is shown as surface. The N-terminal helices of ND2 are shown as cartoons. NDUFC2 and NDUFC1 are shown as cartoons. Lipids are shown as spheres colored by element. (**D**) TMH10$^{ND2}$, TMH11$^{ND2}$, and TMH10-11$^{ND2}$ loop of *D. melanogaster* and *O. aries* (PDB:6ZKC) are shown as cartoons. (**E**) ND2-NDUFA10 interface in *D. melanogaster* (this study) and *O. aries* (PDB:6ZKC) is shown. ND2, NDUFA10 are shownas surface. TMH10$^{ND2}$ TMH11$^{ND2}$ and TMH10-11$^{ND2}$ loop are shown as cartoons. Lipids are shown as spheres colored by the element. (**F**) NDUFA11 in *Y. lipolytic* (PDB: 6YJ4) (blue), *D. melanogaster* (this study) (orange), and *O. aries* (PDB:6ZKC) (yellow) is shown as cartoons.

The online version of this article includes the following figure supplement(s) for figure 2:

**Figure supplement 1.** Structural analysis of NDUFS1-NDUFA2 interface.

**Figure supplement 2.** Structural analysis of NDUFS7.

**Figure supplement 3.** Structural analysis of ND2-NDUFA10 interface.

**Figure supplement 4.** NDUFA11.

*supplement 1C*). Sequence alignment revealed that this loop is four residues shorter compared to *Y. lipolytica* and mammals and that several bulky residues have been replaced by alanine or glycine residues in *Dm*-NDUFS1 (*Figure 2—figure supplement 1D*) resulting in the loss of ordered secondary structure and increased flexibility. The helical structure in the NDUFS1 loop is not seen in bacterial CIs (*Baradaran et al., 2013*; *Kolata and Efremov, 2021*; *Kravchuk et al., 2022*), suggesting that this secondary structure element may have evolved in eukaryotes specifically to interact with NDUFA2.

In mammals, NDUFA2 plays an important role in the stability of the N-module (*Stroud et al., 2016*) and has been proposed to play a role in the regulation of CI by ROS (*Padavannil et al., 2021*). Whereas NDUFA2 knockout in HEK293T cells results in the loss of the N-module and complete loss of CI activity (*Stroud et al., 2016*), NDUFA2 knock-down in *D. melanogaster* has minimal effects, with CI retaining ~97% of WT (Mhc-Gal4/W$^{1118}$ flies) activity during the first week after eclosing as adults (*Garcia et al., 2017*). It would be interesting to explore how a CRISPR-mediated knockout of NDUFA2 affects CI activity in *D. melanogaster*.

Given that NDUFA2 only interacts with NDUFS1, it was proposed that NDUFA2 binding compensates for truncation of NDUFS1's 'D domain' that is otherwise present in bacterial orthologs (*Figure 2—figure supplement 1B, C*; *Padavannil et al., 2021*). However, like other eukaryotes, domain D remains short in *Dm*-NDUFS1 and the reason for the sustained stability of *Dm*-CI when NDUFA2 is knocked down is unclear. Additional studies using a CRISPR-mediated knockout strain of NDUFA2 should help resolve this conundrum.

*NDUFS7* – As in *Y. lipolytica*, NDUFS7 in *D. melanogaster* has an extended N-terminus relative to mammalian CI (*Figure 2B*, *Figure 2—figure supplement 2A, B*). However, whereas the N-terminus of *Yl*-NDUFS7 binds along the surface of NDUFA9, that of *Dm*-CI is flipped ~180° binding overtop of NDUFA12 (*Figure 2B*, *Figure 2—figure supplement 2A*). This additional interface would stabilize the association of NDUFA12 in *Dm*-CI. During CI assembly, the assembly factor NDUFAF2 binds at the equivalent position of NDUFA12 (*Parey et al., 2019*) and is exchanged for NDUFA12 before the full assembly of the PA (*Andrews et al., 2013*;

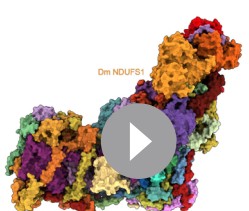

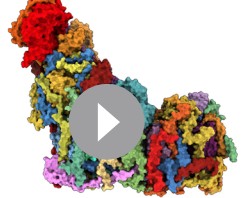

**Video 3.** Structural analysis and comparison of NDUFS1-NDUFA2 interface in *Y. lipolytica* (PDB:6YJ4), *D. melanogaster* (this study), and *O. aries* (PDB:6ZKC). https://elifesciences.org/articles/84415/figures#video3

**Video 4.** Structural analysis of ND2, comparison of ND2-NDUFA10 interface and comparison of ND2-NDUFC2 interface in *D. melanogaster* (this study) and *O. aries* (PDB:6ZKC). https://elifesciences.org/articles/84415/figures#video4

*Vogel et al., 2007*). The additional interactions between NDUFS7 and NDUFA12 in *Dm*-CI may thus influence the assembly of the PA by promoting the exchange of NDUFAF2 with NDUFA12 through the stabilization of NDUFA12 binding (*Figure 2B*).

*NDUFC1* – As in mammalian CI, *Dm*-ND2 lacks three TMHs at the N-terminus, thus having 11 TMHs as opposed to the 14 TMHs otherwise seen in bacteria, plants, ciliates, and yeast (*Birrell and Hirst, 2010*; *Figure 2C* and *Figure 2—figure supplement 3A*). The cavity formed by the lack of the three TMHs is filled with lipids that are held in place in part by the NDUFC2 subunit (*Figure 2C*). Similar to other eukaryotic CIs, the last TMH in *Dm*-ND2 (TMH11$^{ND2}$), has two additional turns, compared to TMH11$^{ND2}$ of mammals (*Figure 2D* and *Figure 2—figure supplement 3B*; *Klusch et al., 2021*; *Parey et al., 2019*; *Zhou et al., 2022*). It has been proposed that in mammals, subunit NDUFC1 plays a role in shortening TMH11$^{ND2}$ by binding a cardiolipin molecule that caps the helix stabilizing its partially unwound state (*Padavannil et al., 2021*). Given that *D. melanogaster* lacks accessory subunit NDUFC1 and has a longer TMH11$^{ND2}$ relative to mammals, this supports the proposed role of NDUFC1 in mammals. Lack of NDUFC1 in *Dm*-CI also indicates that it was recruited as an accessory subunit only after the split of Protostomia and Deuterostomia.

In both mammals and *D. melanogaster*, the TMH10-11$^{ND2}$ loop provides a major interface with the metazoan-specific subunit NDUFA10 (*Figure 2E* and *Video 4*). In *Y. lipolytica*, which lacks any subunit binding on the matrix side of ND2, as well as in plants and Tetrahymena which use the equivalent loop to interact with their γ-carbonic anhydrase subunit (*Klusch et al., 2021*; *Maldonado et al., 2020*; *Soufari et al., 2020*; *Zhou et al., 2022*), the TMH10-11$^{ND2}$ loop spans across the matrix surface of ND2 as a coil. In mammals, TMH10-11$^{ND2}$ spans the same distance across the matrix surface but forms a short α-helix that interacts directly with NDUFA10 (*Figure 2D and E* and *Video 4*). Given that the length of the TMH10-11$^{ND2}$ loop is only shorter by two residues in *D. melanogaster* compared to mammals (*Figure 2—figure supplement 3B*), the additional residues required for the TMH10-11$^{ND2}$ loop to fold into an α-helix in mammals must come from the unwinding of TMH11$^{ND2}$. Thus, a simple model emerges for how mammalian CI is dependent on NDUFC1 for assembly. Namely, the mammalian interface between ND2 and NDUFA10 cannot form before NDUFC1 binds and recruits a cardiolipin to partially unwind TMH11$^{ND2}$ (*Figure 2—figure supplement 3C*; *Padavannil et al., 2021*). This model is consistent with the known binding order and dependencies of NDUFC1 and NDUFA10 to ND2 during CI assembly (*Guerrero-Castillo et al., 2017*; *Stroud et al., 2016*).

*NDUFA11* – In *Dm*-CI, due to an extended C-terminal coil, the MA accessory subunit NDUFA11 has much more extensive interaction with the core subunits than seen in mammals (*Figure 2F*, *Figure 2—figure supplement 4A and B* and *Video 5*). NDUFA11 is a four-TMH subunit that binds adjacent to ND2 atop the ND5 lateral helix (ND5-HL) and ND5 TMH15 (TMH16 in fungi and mammals). It has an arch shape with a concave surface facing CI. The cavity formed at the interface is filled with lipids that bridge between NDUFA11 and ND2/ND5. In mammalian CI, only limited protein-protein contacts between NDUFA11 and other CI subunits are observed, dominated mainly by its short C-terminal coil (*Figure 2F* and *Figure 2—figure supplement 4A*). For this reason, detergent extraction can result in the loss of NDUFA11 in mammals, resulting in so-called 'state 3' particles (*Chung et al., 2022b*; *Fiedorczuk et al., 2016*; *Zhu et al., 2016*). However, in the fungus *Y. lipolytica* the C-terminus of NDUFA11 is much longer and extends along the matrix side of the complex (*Parey et al., 2019*). Likely for this reason NDUFA11 in *Y. lipolytica* is harder to dissociate from the complex even after treatment with harsh detergents (*Angerer et al., 2011*).

Like *Y. lipolytica*, NDUFA11 in *Dm*-CI has an extended C-terminus (*Figure 2F* and *Figure 2—figure supplement 4A*). However, unlike the C-terminus in *Y. lipolytica* NDUFA11 which runs along the length of the MA, that of *Dm*-NDUFA11 runs across the membrane arm (i.e. perpendicular to the long axis), tucking between NDUFB5, NDUFS5, NDUFA8, and ND2, emerging adjacent to NDUFC2 on the opposite side (*Figure 2F* and

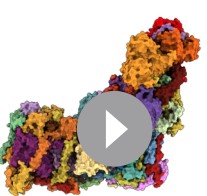

**Video 5.** Structural analysis of NDUFA11 subunit of *Dm*-CI and structural comparison of NDUFA11 in *Y. lipolytica* (PDB:6YJ4), *D. melanogaster* (this study), and *O. aries* (PDB:6ZKC).

https://elifesciences.org/articles/84415/figures#video5

*Video 5*). This arrangement is not seen in any other known CI structure. The C-terminus of *Dm*-NDUFA11 occupies space that is occupied by subunit NDUX1 (NUXM) in yeast, plants, and ciliates (a subunit that is lost in metazoans along with the truncation of ND2), and the N-terminus of NDUFC2 in mammals (*Fiedorczuk et al., 2016*; *Klusch et al., 2021*; *Maldonado et al., 2020*; *Parey et al., 2019*; *Zhou et al., 2022*). The extended C-terminus of *Dm*-NDUFA11 has significant implications for the assembly of *Dm*-CI. Mammalian CI is assembled through a series of intermediates and NDUFA11 is a terminally associated protein that does not form a part of any assembly intermediate (*Guerrero-Castillo et al., 2017*). Also in mammals, NDUFB5, NDUFB8, and NDUFS5 on the IMS side are all present in assembly intermediates prior to the addition of NDUFA11 (*Guerrero-Castillo et al., 2017*). The arrangement seen in *Dm*-CI, in which the C-terminus of *Dm*-NDUFA11 is sandwiched between *Dm*-NDUFB5, *Dm*-NDUFA8, *Dm*-NDUFS5, and *Dm*-ND2 (*Video 5*), suggests that *Dm*-NDUFA11 would need to bind *Dm*-ND2 before *Dm*-NDUFB5, *Dm*-NDUFA8, and *Dm*-NDUFS5. However, as NDUFA11 binds on top of ND5-HL, ND5 would need to associate with ND2 before NDUA11 could bind. This order of events is counter to what occurs in mammals (*Guerrero-Castillo et al., 2017*) and indicates that like plant CI (*Ligas et al., 2018*), assembly of this region of *Dm*-CI might proceed via a distinct mechanism than that of mammals.

## Features of *Dm*-CI subunits with implications for regulation

*NDUFV3* – In mammals there are two isoforms of the NDFUV3 subunit – a long isoform NDUFV3-L and a short isoform NDUFV3-S – and it has been proposed that binding of the different isoforms may impact the activity of CI (*Bridges et al., 2016*; *Dibley et al., 2017*; *Guerrero-Castillo et al., 2016*). Density for a subunit at the position of NDUFV3 in mammals was also observed in the *Dm*-CI structure (*Figure 3A*). However, in the average structure calculated using all *Dm*-CI particles, this density was weak compared to that of the surrounding core subunits. When focused refinements were performed with a mask around the tip of the PA, two clear classes could be isolated differing in the occupancy of the NDUFV3 site (*Figure 3A* and *Figure 3—figure supplement 1*). Thus, in *Dm*-CI this site is only partially occupied.

The 68 kDa fragment of the atypical cadherin (Ft4) can regulate *Dm*-CI activity (*Sing et al., 2014*) and it was proposed that it may bind to *Dm*-CI at the NDUFV3 site (*Bridges et al., 2016*). The density of the occupied class was too noisy to confidently assign the sequence of the subunit from the map (*Figure 3A*). However, our results are consistent with the hypothesis that binding at this site may be regulatory (*Bridges et al., 2016*).

*NDUFA10 and NDUFA5* – In addition to the major interface between NDUFA10 and ND2, in mammals NDUFA10 forms a state-dependent interface with accessory subunit NDUFA5 (*Figure 3—figure supplement 2A*; *Agip et al., 2018*; *Kampjut and Sazanov, 2020*; *Letts et al., 2019*). In *Dm*-CI, the interface between NDUFA10 and NDUFA5 is larger due to an extended NDUFA10 N-terminal coil that inserts between NDUFA10 and NDUFA5 (*Figure 3B and C* and *Figure 3—figure supplement 2B*). It has been debated whether the breaking of the interface between NDUFA10 and NDUFA5 occurs during enzyme turnover or is a feature of the D state (*Agip et al., 2018*; *Kampjut and Sazanov, 2020*). The enhanced interface between *Dm*-NDUFA10 and *Dm*-NDUFA5 would make these subunits more difficult to separate and indicates that, although the interactions at this interface may be state-dependent (see below), the interface is less likely to be fully broken as seen in the mammalian context (*Agip et al., 2018*; *Kampjut and Sazanov, 2020*; *Letts et al., 2019*).

## Features of *Dm*-CI with implications for SC assembly

The structure provides a basis for understanding the lower abundance of SCs between *D. melanogaster* and mammals. In mammals, the N-terminus of NDUFB4 forms part of the only matrix interface between CI and CIII$_2$ in SC I+III$_2$ (*Letts et al., 2019*; *Letts and Sazanov, 2017*). In this interaction, a loop from the UQRC1 subunit of one CIII protomer binds in between the N-terminus of NDUFB4 and the three-helix-bundle of subunit NDUFB9. In *Dm*-CI the N-terminus of *Dm*-NDUFB4 is truncated relative to that of mammals and does not extend far enough towards NDUFB9 to form this interface (*Figure 3D* and *Figure 3—figure supplement 3*). Thus, the lack of this interface likely contributes to the observed low abundance of SCs in *D. melanogaster* mitochondria.

Additionally, given its role in bridging between CI and CIII$_2$ in respiratory SCs (*Letts et al., 2019*; *Letts and Sazanov, 2017*), it has been proposed that NDUFA11's interaction with CI may have been

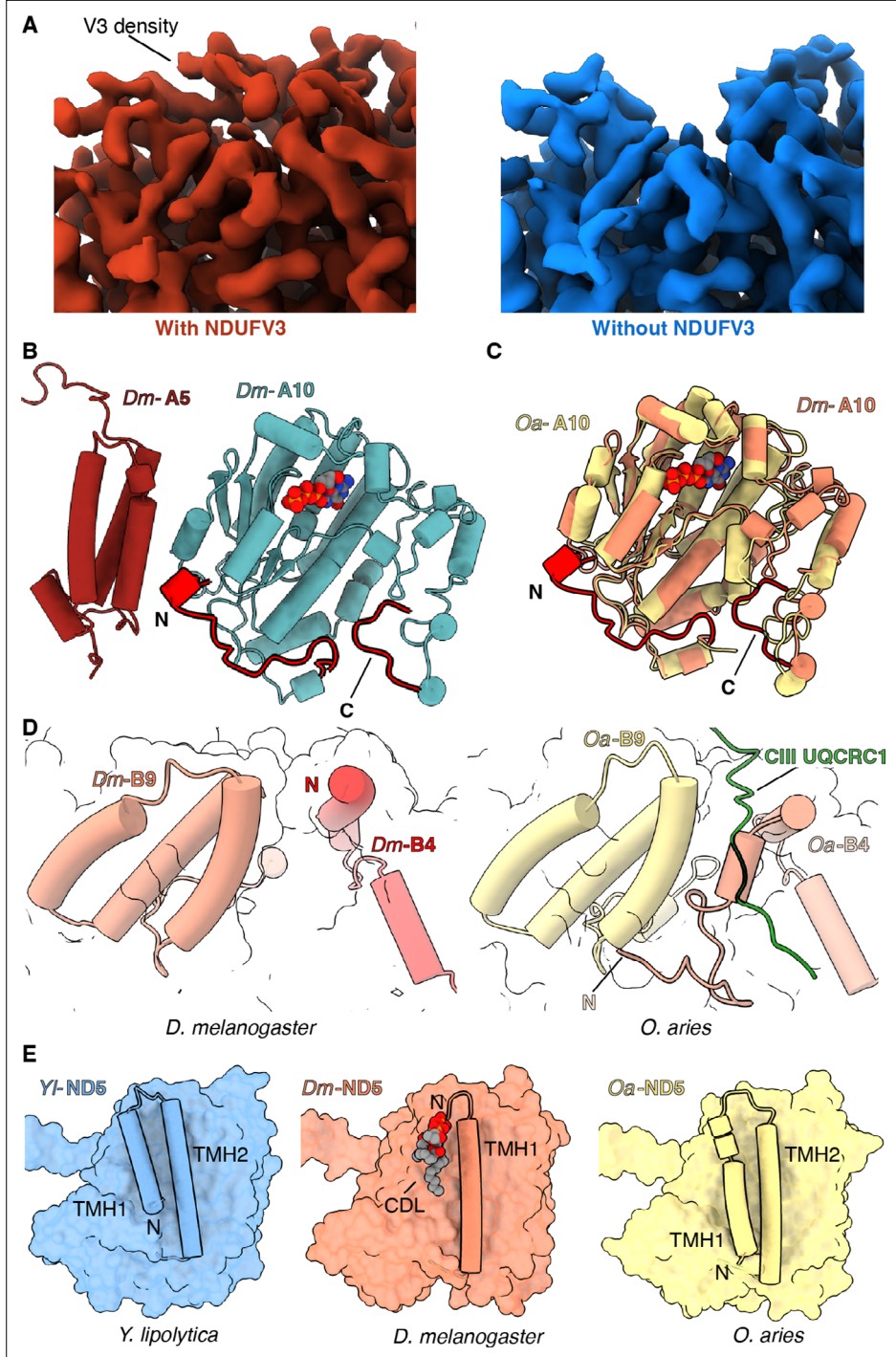

**Figure 3.** Features of *Dm*-CI subunits that may impact regulation, SC formation, or lipid binding. (**A**) Cryo-EM map with (red) and without (blue) NDUFV3 density is shown. (**B**) NDUFA10-NDUFA5 interface in *Dm*-CI is shown. The subunits are shown in cartoons colored as in *Figure 1*. The N and C terminal extension of *Dm*-NDUFA10 are colored red. (**C**) Structural alignment of NDUFA10 from *O. aries* (yellow) (6ZKC) and *D. melanogaster* (orange) is shown. The N and C-terminal extensions of *Dm*-NDUFA10 are shown in red. (**D**) N-terminal region of NDUFB4 and NDUFB9 in *D. melanogaster* (this study) and *O. aries* (PDB:6QC3) is shown as cartoons. Loop region of *Oa*-CIII subunit UQCRC1 forming interface with *Oa*-CI subunit is shown as cartoon colored in green. (**E**) ND5 in *Y. lipolytic* (PDB: 6YJ4), *D. melanogaster* (this study) and *O. aries* (PDB:6ZKC) is shown as surface. The N-terminal helices of ND5 are shown as cartoons. Lipids are shown as spheres colored by the element.

The online version of this article includes the following figure supplement(s) for figure 3:

*Figure 3 continued on next page*

*Figure 3 continued*

**Figure supplement 1.** NDUFV3 is sub-stoichiometric.

**Figure supplement 2.** State-dependent interactions between NDUFA5 and NDUFA10.

**Figure supplement 3.** NDUFB4.

**Figure supplement 4.** *Dm*-ND5 has a truncated N-terminus.

**Figure supplement 5.** NDUFC2.

---

weakened in species, such as mammals, to promote SC formation by requiring the presence of CIII$_2$ to hold NDUFA11 in place (**Padavannil et al., 2021**). The extended interaction interface provided by the C-terminus of *Dm*-NDUFA11 indicates that *Dm*-NDUFA11 is not dependent on the presence of CIII$_2$ for its stable association with the *Dm*-CI. Thus, the *Dm*-CI structure supports the proposed inverse relationship between the strength of the interaction between NDUFA11 and CI and the abundance of SCs (**Padavannil et al., 2021**).

## Differences in lipid binding

Lipids form an integral part of the CI MA and loss of lipids during purification results in diminished CI activity (**Letts et al., 2016a**; **Padavannil et al., 2021**). Across species, structural lipids are seen tightly binding to the surface of CI at the interface of the core H$^+$-pumping subunits and specific deformation of the lipid membrane by NDUFA9 occurs adjacent to the Q-tunnel (**Kampjut and Sazanov, 2020**; **Parey et al., 2019**; **Zhou et al., 2022**). In general, the pattern of lipid binding to *Dm*-CI (**Figure 1— figure supplement 4I**) is like that seen in other CI structures with two notable exceptions. First, instead of 16 TMHs observed in ND5 of *Y. lipolytica* and mammals, *Dm*-ND5 lacks the first TMH for a total of only 15 (**Figure 3E** and **Figure 3—figure supplement 4A**). Like what is seen with the metazoan-specific shortening of ND2 discussed above, the region occupied by ND5-TMH1 in other species is not occupied by any other protein subunit but binds lipid (**Figure 3E**, **Figure 3—figure supplement 4A**). As mammalian ND5 maintains its full complement of helices, loss of the first TMH must have occurred after the split of Protostomia and Deuterostomia, but how widespread the ND5 deletion is in Protostomia along with any functional implications remains to be determined.

Second, in mammals, the N-terminus of the two-TMH accessory subunit NDUFC2 binds underneath ND2 in the pocket left by the deletion of the first three ND2 TMHs. However, in *Dm*-CI this space is filled by the C-terminus of NDUFA11 (**Figure 2C** and **Video 5**). Like mammalian NDUFC2, *Dm*-NDUFC2 has an extended N-terminus relative to that of *Y. lipolytica* (**Figure 2C** and **Figure 3— figure supplement 5A**); however, instead of crossing over TMH2$^{C2}$ and binding under ND2, it crosses over the C-terminal coil of NDUFA8 and forms additional interactions with NDUFB1 and NDUFB5 (**Figure 2C** and **Figure 3—figure supplement 5A**). These additional interactions contribute to lipid binding at the ND2/ND4 interface by capping this pocket and likely help to stabilize lipid binding at the interface between the two H$^+$-pumping core subunits.

## Focused classification of *Dm*-CI reveals an NDUFS4-helix-locked state

Initial poor resolvability of the average *Dm*-CI map around NDUFA10 at the interface of the PA and MA led us to perform focused classifications using a mask encompassing subunits NDUFA10, NDUFA5, NDUFA6, and NDUFAB1-α (**Figure 4—figure supplement 1**). This classification revealed two major classes and two minor classes of *Dm*-CI particles that differ in the presence of an ordered α-helical element from NDUFS4 bound at the interface of NDUFA5, NDUFA10, and NDUFA6 and in the angle between the PA and MA (**Figure 4**). Only in one of the major classes is the NDUFS4 helix visible, while in the other major class and two minor classes this helix is disordered. We name the states the NDUFS4-helix-locked (helix-locked) state and the flexible class 1-3 states (**Figure 4** and **Figure 4—figure supplement 1**).

The name helix-locked state refers to the fact that the positions of the PA and MA are 'locked' relative to each other due to the presence of the NDUFS4 helix (**Figure 4A**). In the absence of the lock helix, the relative positions of PA and MA vary and thus the complex is more flexible (**Kampjut and Sazanov, 2020**; **Kravchuk et al., 2022**; **Parey et al., 2021**) as indicated by the three distinct subclasses (**Figure 4—figure supplement 1**). Three-dimensional variability analysis (3DVA) performed on

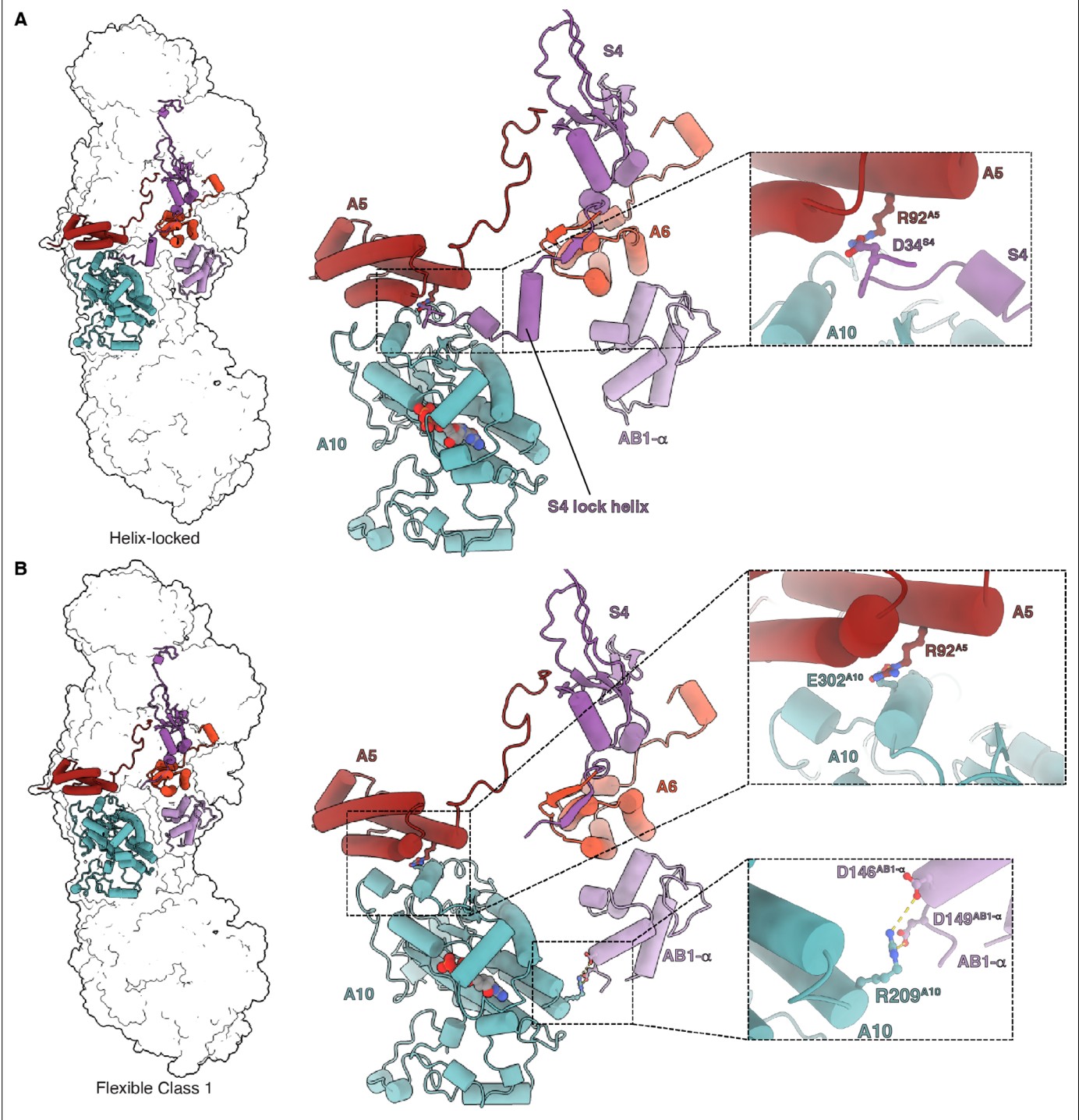

**Figure 4.** Matrix interactions in the helix-locked and flexible class 1 states of *Dm*-CI. (**A**) *Dm*-CI subunits forming the bridge at the peripheral arm (PA)/membrane arm (MA) interface in the helix-locked state are shown as cartoons colored as in *Figure 1*. The interaction between NDUFA5 and the N-terminal region of NDUFS4 is shown in the inset. (**B**) *Dm*-CI subunits forming the hinge at the PA/MA interface in the flexible class 1 state are shown as cartoons colored as in *Figure 1*. NDUFA5, NDUFA10, and NDUFA10, NDUFAB1-α interactions are shown in insets.

The online version of this article includes the following figure supplement(s) for figure 4:

**Figure supplement 1.** Identification of helix-locked state.

**Figure supplement 2.** NDUFS4 comparison.

**Figure supplement 3.** Comparison of *Dm*-CI states.

*Figure 4 continued on next page*

*Figure 4 continued*

**Figure supplement 4.** Comparison of helix-locked state and flexible class 1 state from *Dm*-CI to states of CI from representative organisms.

**Figure supplement 5.** Comparison of relative positions of NDUFA10 and NDUDAB1-α in different CI states.

the full set of particles revealed a smooth transition between the flexible sub-states along with the disappearance of the NDUFS4 helical density (*Video 6*). This indicates that the frozen *Dm*-CI particles adopt conformations along the trajectory from the major flexible class 1 state, which is the furthest rotated relative to the helix-locked state, towards the helix-locked state. Once the PA and MA are positioned such that they are far enough apart to accommodate the NDUFS4 helix, the binding of the helix 'locks' their relative positions (*Figure 4A* and *Video 6*). The N-terminal region of NDUFS4 that forms the lock helix is not conserved in *Y. lipolytica* or mammals (*Figure 4—figure supplement 2*) suggesting that it evolved after the split of Protostomia and Deuterostomia.

Overall, the helix-locked state and the flexible class 1 state mainly differ from each other by a rotation of the PA and the tilting of subunit ND1 (*Figure 4—figure supplement 3*). When compared to the overall open/deactive and closed/active state structures of mammalian CI, the *Dm*-CI states do not align well (*Figure 4—figure supplement 4*). The rotation and angle between the PA and MA in the *Dm*-CI states do not correspond to those seen in any of the mammalian states but are more like states seen in *E. coli* CI and the hyper thermophilic fungus *Chaetomium thermophilum* (*Figure 4— figure supplement 4*; *Kravchuk et al., 2022*; *Laube et al., 2022a*). However, as has been recently acknowledged (*Kravchuk et al., 2022*), across species there is not a clear one-to-one mapping of the observed states indicating that the relative positions of the PA and MA are not indicative of conserved functional states of the complex as initially proposed (*Kampjut and Sazanov, 2020*). One possible measure to compare states in metazoans is the distance between the center of mass of NDUFAB1-α and that of NDUFA10 (*Figure 4—figure supplement 5*). According to this measure, the helix-locked state is most like the closed/active states (*Figure 4—figure supplement 5*) and the *Dm*-CI flexible class 1 state is most like the open/deactive states of mammalian CI (*Figure 4—figure supplement 5*). Although the density of the NDUFAB1-α subunit is weak due to flexibility, its position in flexible class 1 is consistent with the formation of salt-bridges between residues D146[AB1-α], D149[AB1-α] and R209[A10] (*Figure 4B* and *Video 6*). In plant and Tetrahymena CI, the NDUFAB1-α subunit is involved in bridging interactions between the PA and MA (*Klusch et al., 2021*; *Zhou et al., 2022*); however, to our knowledge, this is the first indication of direct bridging between the PA and MA via NDUFA10 and NDUFAB1-α in any metazoan structure.

Although NDUFA5 and NDUFA10 remain in direct contact in both the helix-locked and the flexible states, there is a state-dependent change in their interaction (*Figure 4* and *Video 6*). In the helix-locked state, the interaction between NDUFA5 and NDUFA10 is mediated in part by the N-terminus of NDUFS4 which binds along the surface of NDUFA10; and D34[S4] forms a salt bridge with R92[A5] (*Figure 4A* and *Video 6*). In the flexible class 1 state, the N-terminus of NDUFS4 is disordered, and NDUFA5 slides along the surface of NDUFA10 such that R92[A5] forms a salt bridge with E302[A10] (*Figure 4B* and *Video 6*). Thus, R92[A5] is used to form salt bridging interactions with NDUFS4 and NDUFA10 in a state-dependent manner (*Figure 4* and *Video 6*).

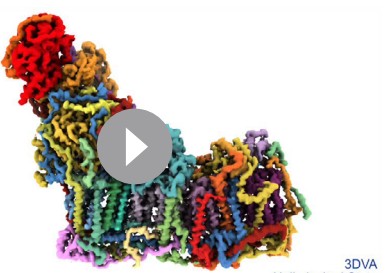

**Video 6.** 3D variability analysis of *D. melanogaster* CI, component 1. The 3DVA volumes are shown as a continuous movie. The movie emphases on the hinge region of peripheral arm (PA)/membrane arm (MA) interface. *Dm*-CI subunits are colored as in *Figure 1*.
https://elifesciences.org/articles/84415/figures#video6

## In the helix-locked state the CoQ reduction site loops are buried by an NDUFA9 'latch'

Conformational changes in loops adjacent to the CoQ reduction site were seen between the helix-locked and flexible class 1 states of *Dm*-CI (*Figure 5* and *Video 7*). In mammals, *Y. lipolytica* and *E. coli* the open state of the complex is commonly associated with specific conformations or disorder of loops around the CoQ binding site (α1-2[S7] loop, α2-β1[S7] loop, TMH5-6[ND1] loop, TMH1-2[ND3] loop, β1-2[S2] loop, and TMH3-4[ND6]

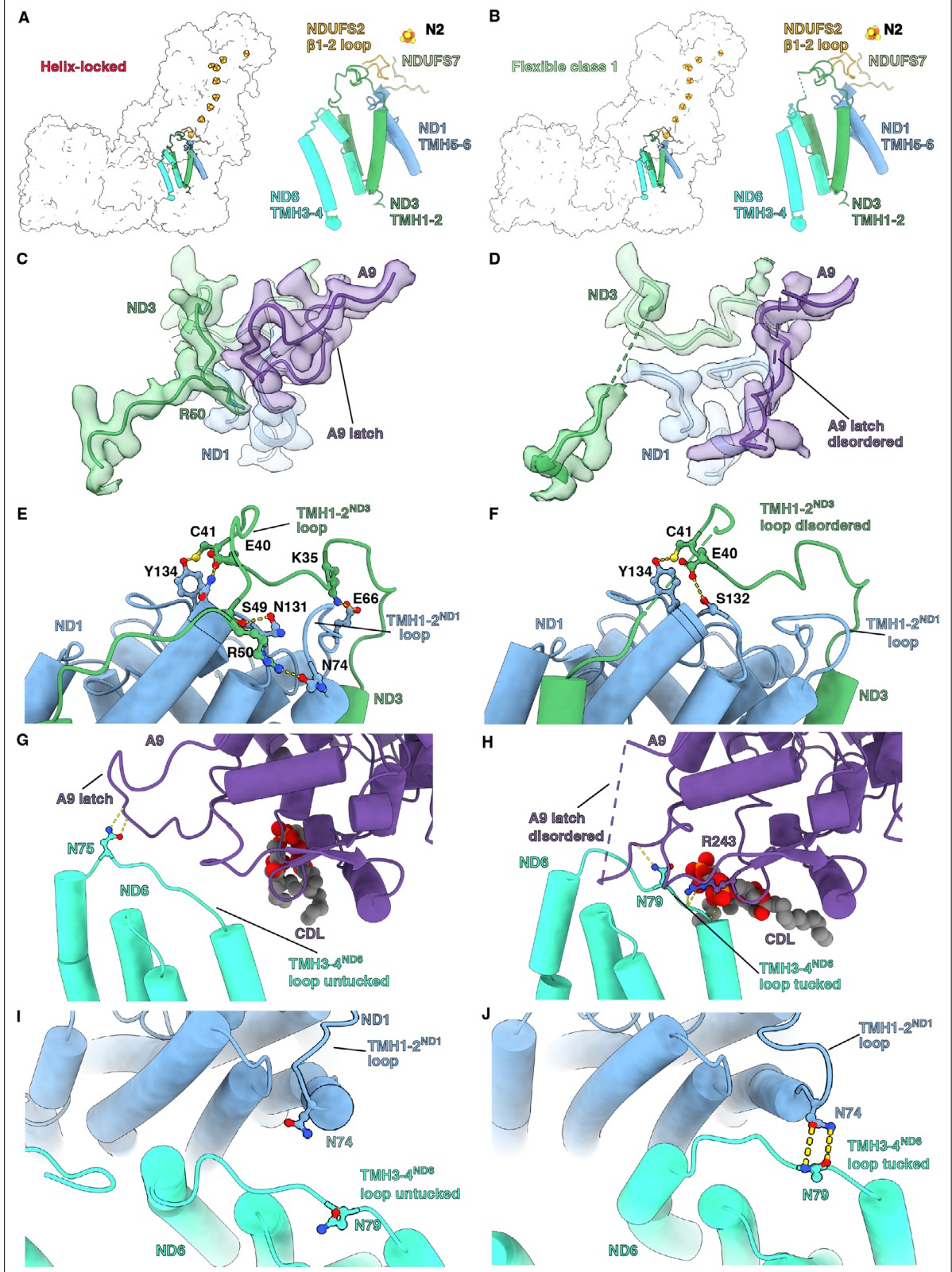

**Figure 5.** Q-site adjacent loops at the peripheral arm (PA)/membrane arm (MA) interface. (**A, B**) The Q-site adjacent loops in the (**A**) helix-locked state and (**B**) flexible class 1 state are shown as cartoon colored as in *Figure 1*. (**C, D**) Interface between ND3, ND1, and NDUFA9 in the (**C**) helix-locked and (**D**) flexible class 1 state is shown. ND3, ND1, and NDUFA9 are shown as colored cartoons embedded in density colored as in *Figure 1*. (**E, F**) ND1 interaction with TMH1-2^ND3 loop in the (**E**) helix-locked and (**F**) flexible class 1 states are shown. ND1 and ND3 are shown as cartoons colored as in

*Figure 5 continued on next page*

*Figure 5 continued*

*Figure 1*. (**G, H**) TMH3-4[ND6] loop interaction with NDUFA9 in the (**G**) helix-locked and (**H**) flexible class 1 states are shown. ND6 and NDUFA9 are shown as cartoons colored as in *Figure 1*. (**I, J**) Interactions at the interface of ND1 and ND6 in the (**I**) helix-locked and (**J**) flexible class 1 states are shown. ND1 and ND6 are shown as cartoons colored as in *Figure 1*. Figure-figure supplements.

The online version of this article includes the following figure supplement(s) for figure 5:

**Figure supplement 1.** Structural comparison of the Q-site adjacent loops of *Dm*-CI in the helix-locked and flexible class 1 states with the confirmations of *O.aries, E. coli,* and *C. thermophilum* CI.

**Figure supplement 2.** Comparison of TMH4[ND6] orientation in the open and closed states across different organisms.

loop) as well as a π-bulge in TMH3[ND6] (*Kampjut and Sazanov, 2020*; *Kravchuk et al., 2022*; *Parey et al., 2021*; *Parey et al., 2018*). In the closed state, these loops are generally well ordered and TMH3[ND6] re-folds into an α-helix (*Figure 5—figure supplement 1*). No differences in the conformations of the α1-2[S7], TMH5-6[ND1], and β1-2[S2] loops were seen in *Dm*-CI between the helix-locked and flexible class 1 states (*Figure 5—figure supplement 1A-C*). The TMH1-2[ND3] loop is well ordered in the helix-locked state and partially disordered, though to a lesser extent than seen in most CI open states, in the flexible class 1 state; and TMH3[ND6] is α-helical in the helix-locked state and contains a π-bulge in the flexible class 1 state (*Figure 5A and B* and *Figure 5—figure supplement 1D and E*). These differences in the TMH1-2[ND3] loop stem from a state-dependent interaction with the C-terminal loop of NDUFA9, a rotation of ND1 relative to the rest of the MA, and the movement of TMH4[ND6] and the TMH3-4[ND6] loop (*Figure 5C–J Video 7*, and *Figure 5—figure supplements 1 and 2*).

In the helix-locked state, the C-terminal loop of NDUFA9 binds atop the TMH1-2[ND3] loop, trapping R50[ND3] to the surface of ND1 (*Figure 5C*), thereby holding the loop in place like a latch. In the flexible class 1 state, the C-terminal loop of NDUFA9 moves away from the TMH1-2[ND3] loop, providing space for conformational flexibility and both loops become partially disordered (*Figure 5D*). Specifically, clear density is lost for R50[ND3] and surrounding residues (*Figure 5D*). The movement of NDUFA9 away from the TMH1-2[ND3] loop is accompanied by changes in the interactions between ND1 and the TMH1-2[ND3] loop caused by the rotation of ND1 and conformational changes in the TMH1-2[ND1] loop. In the helix-locked state, multiple hydrogen bonding and salt bridging interactions were seen between the TMH1-2[ND3] loop and ND1 (*Figure 5E*). However, in the flexible class 1 state most of these interactions are lost except for a hydrogen bond between the conserved C41[ND3] and Y134[ND1] (*Figure 5E and F*). E40[ND3] swaps hydrogen bonding partners from N133[ND1] in the helix-locked state to S132[ND1] in the flexible class 1 state (*Figure 5E and F*). Fewer hydrogen bonds between ND1 and the TMH1-2[ND3] loop would also contribute to the observed higher flexibility of this loop in the flexible class 1 state (*Figure 5C, D, E and F*). Finally, the TMH3-4[ND6] loop moves relative to NDUFA9, going from exposed or 'untucked' in the helix-locked state to 'tucked' under NDUFA9 in the flexible class 1 state (*Figure 5G and H* and *Video 7*). The movement of the TMH3-4[ND6] loop brings it into contact with the TMH1-2[ND1] loop in the flexible class 1 state and requires an ~10 Å translation of TMH4[ND6] (*Figure 5I and J*, *Figure 5—figure supplement 2* and *Video 7*). Similar 'latching' behavior of the NDUFA9 C-terminus and 'tucking' of the TMH3-4[ND6] loop, though different in the structural details, was also recently seen in the structure of the thermophilic yeast *Chaetomium thermophilum* (*Laube et al., 2022b*), suggesting that this may be a general mechanism for regulating CI activity.

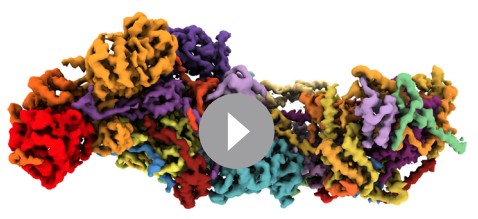

**Video 7.** 3D variability analysis of *D. melanogaster* CI, component 1. The 3DVA volumes are shown as a continuous movie. The movie emphasizes the Q-site and interface loops at the peripheral arm (PA)/membrane arm (MA) interface. *Dm*-CI subunits are colored as in *Figure 1*.
https://elifesciences.org/articles/84415/figures#video7

## Discussion

We present here the structure of mitochondrial CI from the model organism *D. melanogaster* (*Dm*-CI), providing a representative structure for insects and Protostomes more generally – a broad group of animals that split from mammals and other Deuterostomes approximately 600 million years ago. During the evolution of metazoans, a split in the biochemical behavior of mitochondrial

CI occurred corresponding to the Protostomia/Deuterostomia divide (*Figure 1A*). According to established biochemical assays (*Babot et al., 2014*), all characterized CIs from Deuterostomes can enter a NEM sensitive deactive 'D' state, a property that they share with fungal CIs, whereas this is not the case for CI from Protostomes (*Figure 1A*; *Babot et al., 2014*; *Maklashina et al., 2003*). However, it is important to note that although all current data are consistent with a Protostomia/Deuterostomia split in the biochemical behavior of CI, only a small subset of species from each group have been investigated (*Figure 1A*). Characterization of additional species across both groups will provide a better understanding of the evolution of this functional split. We show here that although *Dm*-CI is insensitive to NEM after incubation at elevated temperatures in the absence of substrate (*Figure 1B, E and G*, and *Figure 1—figure supplement 1D, F*) it does shows an activation lag phase that is consistent with the presence of an off-pathway resting state (*Figure 1E and G* and *Figure 1—figure supplement 1D, F*). Recently the structural basis of the A-to-D transition has become central to the debate over the CI coupling mechanism (*Chung et al., 2022a*; *Kampjut and Sazanov, 2022*).

In fungi and mammals, the state-dependent accessibility of the ND3 cysteine in the active 'A' and D states is clearly understood from CI structures. The mammalian CI structure has been solved in multiple states that broadly fall into two categories, either 'open' or 'closed,' originally defined by the angle between the PA and the MA (*Agip et al., 2018*; *Kampjut and Sazanov, 2020*; *Parey et al., 2021*). In the open state the TMH1-2$^{ND3}$ loop, which harbors the reactive cysteine, is disordered indicating flexibility and accessibility, whereas in the closed state, the TMH1-2$^{ND3}$ loop is well ordered and the reactive cysteine is buried and inaccessible. This led to the proposal that the open state of CI corresponds to the D state and the closed state of the complex corresponds to the A state (*Agip et al., 2018*; *Zhu et al., 2016*). This proposal is supported by the structure of deactivated bovine CI which was found to contain 87.5% open-state particles (*Blaza et al., 2018*). However, it was also found that under turnover conditions ovine CI adopts open states, leading Kampjut and Sazanov to propose that open states are part of the catalytic cycle and that the deactive state is a particular 'deep' open state (*Kampjut and Sazanov, 2020*). They tested their model by deactivating the ovine complex prior to structure determination and reporting a large conformational shift in TMH4$^{ND6}$ (*Figure 5—figure supplement 2*; *Kampjut and Sazanov, 2020*). Thus, they conclude that the deactive state is a specific open state characterized by a large displacement of TMH4$^{ND6}$ (*Kampjut and Sazanov, 2022*; *Kampjut and Sazanov, 2020*; *Kravchuk et al., 2022*). However, all deposited maps of the deactivated ovine complex (EMDB-11260, EMDB-11261, EMDB-11262, and EMDB-11263) show very weak or no density for ND5-HL, TMH16$^{ND5}$, NDUFA11 or TMH4$^{ND6}$ (*Kampjut and Sazanov, 2020*). Poor density in these regions has been associated with so-called 'state 3' particles which are proposed to correspond to CI in the first stages of dissociation (*Chung et al., 2022b*; *Zhu et al., 2016*). This leads to the possibility that, despite being able to measure activity, the deactivation treatment of Kampjut and Sazanov may have partially denatured the ovine complex or sensitized it to disruption during the cryoEM grid preparation (*Kampjut and Sazanov, 2020*). If these structures represent the D state, then it follows that disorder of ND5-HL, TMH16$^{ND5}$, and NDUFA11 in addition to the movement of TMH4$^{ND6}$ would also be features of the D state. Although that is possible, despite lower resolution, these features were not observed in the deactivated bovine complex (EMDB-3731) (*Blaza et al., 2018*). Altogether, it is clear that CI's deactive state is an open state. However, given the discrepancies between mammalian structures, it remains unclear how the deactive open state differs from other open states that may be part of CI's catalytic cycle.

In the 'open-and-closed-states' model of CI turnover, the proposed function of open states during catalysis is twofold. Firstly, the disordering of the CoQ-site loops, in particular the TMH1-2$^{ND3}$ and β1-2$^{S2}$ loops, would disrupt the CoQ binding site and open the CoQ-binding cavity to solvent after the formation of ubiquinol (CoQH$_2$), thereby 'washing out' the highly hydrophobic substrate that may otherwise remain 'stuck' in the active site tunnel. Second, an open state during turnover would disrupt the hydrophilic axis via the formation of a π-bulge in TMH3$^{ND6}$, rotating hydrophobic residues into the axis and preventing a futile cycle caused by the 'back flow' of protons to the solvent-accessible CoQ site (*Kampjut and Sazanov, 2022*; *Kampjut and Sazanov, 2020*; *Kravchuk et al., 2022*).

Alternatively, the 'closed-states-only' model proposes that catalysis only occurs through a series of closed states and that all observed open states correspond to the D state (*Chung et al., 2022a*; *Chung et al., 2022b*). In this model, the multiple observed open states would stem from greater flexibility between the PA and MA caused by the angle between them increasing and the release of

the TMH1-2$^{ND3}$ loop from the PA/MA interface and all open states observed from structures under turnover conditions are artifacts of sample preparation or the harsh conditions of the cryoEM grid. In the closed-states-only model, CoQ entry and exit would occur though standard diffusion in and out of the active site tunnel with more subtle conformational changes resulting in changing affinities for CoQ and CoQH$_2$ (*Chung et al., 2022b*; *Chung et al., 2022a*). Also, this model of turnover proposes that the α-helix-to-π-bulge transition of TMH3$^{ND6}$ does not occur during catalysis and that the TMH3$^{ND6}$ π-bulge is only a feature of the D state. Both models are consistent with conformational changes in the CoQ site loops being important during turnover, which has been demonstrated through site-specific cross-linking of the TMH1-2$^{ND3}$ loop with NDUFS7 (*Cabrera-Orefice et al., 2018*), but differs in the degree of conformational change needed.

The structures of resting *Dm*-CI reported here inform this debate. First, in the helix-locked state, the angle between the PA and MA is not consistent with either the open states of *Y. lipolytica* or mammalian CIs but the active site loops are fully ordered and buried by the NDUFA9 latch (*Figure 5C*). This state also lacks the TMH3$^{ND6}$ π-bulge indicating that the water wire of the hydrophilic access is intact, though a higher resolution structure is needed to confirm this. These features are most consistent with the closed/active state of CI. However, the NDUFA9 latch binding on top of the TMH1-2$^{ND3}$ loop and pinning down Arg50$^{ND3}$ (*Figure 5C*) is a feature not seen in mammalian closed/active states to date. Second, in the flexible class 1 state in which the PA and MA twist to bring NDUFAB1-α and NDUFA10 into close proximity, we see increased flexibility in the TMH1-2$^{ND3}$ loop, and the formation of the TMH3$^{ND6}$ π-bulge is more consistent with the open/deactive state of CI. However, it is important to note that the range of disordered residues in the TMH1-2$^{ND3}$ loop is distinct and shorter than that seen in other open/deactive states (*Figure 5—figure supplement 1*) and the conserved Cys41$^{ND3}$ remains well-ordered and hydrogen-bonded to Tyr134$^{ND1}$ (*Figure 5E and F*). Additionally, unlike what is seen in other species, no differences are seen between the β1-2$^{S2}$ loop, the α1-2$^{S7}$ loop, and the TMH5-6$^{ND1}$ loop between the *Dm*-CI states (*Figure 5—figure supplement 1A, B, C*).

Given that both the helix-locked and flexible class 1 states are resting-state structures, i.e., no substrate was provided to the complex prior to flash freezing, we cannot make strong conclusions regarding the catalytic relevance of these states. Nonetheless, four scenarios are possible: (1) both states are catalytically relevant on-pathway states, (2) the flexible state is on the pathway and the helix-locked state is an off-pathway resting state, (3) the flexible state is an off-pathway resting state and the helix-locked state is on the pathway, or (4) both are off-pathway resting states. From the current structures, we are unable to fully differentiate between these scenarios. However, scenario 1 is unlikely given that we were able to detect the presence of an off-pathway resting state (*Figure 1E and G* and *Figure 1—figure supplement 1D, F*) and the unique features of the *Dm*-CI states. The presence and absence of the NDUFS4 lock-helix, which is not adjacent to the active site nor conserved in other CIs of known structure (*Figure 4—figure supplement 3*), suggests that the NDUFS4 lock-helix is a regulatory element that has evolved in a specific branch of eukaryotes to either stabilize the resting (scenario 2) or active (scenario 3) states. If the NDUFS4 lock-helix evolved to regulate the transition between the off-pathway resting state and on-pathway states this would also argue against scenario 4 in which all the observed states are off-pathway. Further, scenario 1 would necessitate the binding and release of the lock-helix during turnover, which may not be rate limiting on the timescale of CI turnover but seems unlikely given the absence of the lock-helix in other species. Therefore, scenarios 2 and 3 seem most likely, though further work is needed to establish which possible scenario is correct.

The 'opening-and-closing' model of CI turnover would favor scenario 2. Large conformational changes in the TMH1-2$^{ND3}$ loop, such as an order-to-disorder transition, during CI turnover (*Cabrera-Orefice et al., 2018*) would also support scenario 2. Flexibility and opening of the TMH1-2$^{ND3}$ loop during turnover are supported by the enhanced reactivity of the ND3 cysteine seen in *Bos taurus* CI and our 2 mM NEM pre-activation conditions (*Figure 1B*; *Burger et al., 2022*). The presence of the NDUFA9 latch holding the TMH1-2$^{ND3}$ loop down in the helix-locked state may block conformational changes in TMH1-2$^{ND3}$, suggesting that it is a resting state incapable of turnover. The NDUFA9 latch would also explain the insensitivity of this potential off-pathway resting state to NEM (*Figure 1B*). If the *Dm*-CI flexible state is a catalytically competent state, the lack of the NDUFA9 latch would allow for conformational changes in the TMH1-2$^{ND3}$ loop and it would support the hypothesis that the TMH3$^{ND6}$π-bulge forms as part of the catalytic cycle.

Given that the proposed physiological function of the D state is to prevent reactive oxygen species production by reverse electron transport (*Chouchani et al., 2013*), it is possible that multiple distinct mechanisms have evolved to achieve this end. If the helix-locked state is a D-like resting state, it suggests that reverse electron transport by CI can be blocked in two distinct ways: (1) opening the CoQ site to solvent and thereby preventing $CoQH_2$ binding as seen in the standard D state; or (2) blocking conformational changes needed for coupling between CoQ reduction/$CoQH_2$ oxidation and $H^+$-pumping as seen in the NDUFA9-latched-helix-locked-state. If the helix-locked state is an auto-inhibited state, these structures represent a novel regulatory mechanism that may be exploited to inhibit CI turnover in other species.

The 'closed-states-only' model of CI turnover would favor scenario 3. According to this model, only the helix-locked state could be the active state as it contains the fully α-helical $TMH3^{ND6}$ which is proposed to only be found in the π-bulge conformation in D-like resting states. If so, given the presence of the NDUFA9 latch this would suggest that either the $TMH1-2^{ND3}$ loop does not undergo conformational changes during turnover or the conformational changes are concerted across multiple subunits and would require additional open-like-states to explain the evidence that the cysteine on this loop becomes exposed during turnover (*Figure 1B*; *Burger et al., 2022*). Nonetheless, this interpretation is consistent with the structure of *T. thermophila* CI in which the $TMH1-2^{ND3}$ loop is also buried (*Zhou et al., 2022*). Given the presence of the lock-helix, in this scenario, CI would turnover without large changes in the angle between the PA and MA. This would be consistent with species whose CI have additional bridging interactions between the PA and MA that may limit changes in the PA/MA angle, such as plants, *Tetrahymena*, and the thermophilic yeast *Chaetomium thermophilum* (*Klusch et al., 2021*; *Laube et al., 2022b*; *Zhou et al., 2022*).

Recently, structures of *E. coli* CI, which does not deactivate (*Maklashina et al., 2003*) but does have an uncoupled resting-state (*Belevich et al., 2017*), revealed a variety of 'open' states in which the $TMH1-2^{ND3}$ loop was disordered and $TMH3^{ND6}$ featured a π-bulge (*Kravchuk et al., 2022*). In this study, a mixture of open and 'closed' states, in which the $TMH1-2^{ND3}$ loop is well ordered and $TMH3^{ND6}$ is full α-helical, was only observed under turnover conditions (*Kravchuk et al., 2022*). This confirms that the closed state is a catalytically relevant active state and supports the hypothesis that open states are not solely a consequence of deactivation. However, to demonstrate that the open states are catalytically relevant other possible explanations for the observation of open states must be ruled out. Thus, it is important to note that another recent structural study of *E. coli* CI found that intact biochemically active preparations contained a significant fraction of broken particles on the cryoEM grids (*Kolata and Efremov, 2021*). After classification Kolata and Efremov found that most *E. coli* CI particles used for the reconstruction of the PA (151,357 vs. 134,976 particles) were particles that had completely dissociated from the MA (*Kolata and Efremov, 2021*). The biochemical preparations for the *Kravchuk et al., 2022*, and Kolata and Efremov studies are different and Kravchuk et al., used gentler conditions and do not report any disrupted particles. Nonetheless, the results of Kolata and Efremov are a stark reminder that structural studies on extracted membrane protein complexes exist within a spectrum of biochemical stability and that complexes that are intact according to size exclusion chromatography, mass photometry, and activity assays, can still end up as broken particles on the cryoEM grid (*Han et al., 2022*; *Kolata and Efremov, 2021*). Therefore, it is not unreasonable to consider that the interaction with the air-water interface during grid preparation may act to convert closed-state CI to the lower energy (higher entropy due to disorder and flexibility) open states.

Therefore, interactions with the air-water interface should be noted as an alternate explanation as to why most structures of CI across species appear to have open states, and complexes in which the PA and MA are more stably associated should be sought out for additional corroborative functional work. Although *Dm*-CI adopts an open-like flexible state, it may be more resistant to disruption during grid preparation due to the expanded interaction between NADUFA5 and NDUFA10 (*Figure 3B*). However, *Dm*-CI needs to be further characterized in the presence of substrates before general conclusions can be drawn. Due to the presence of the lock-helix, *Dm*-CI is an ideal system for further structural and functional work as the presence of the NDUFS4 helix can act as a strong signal for the different states under different turnover and grid preparation conditions.

In conclusion, the structure of *Dm*-CI from thoracic muscle reveals unique features of CI from Protostomia that do not share the standard A-to-D transition as defined biochemically. Overall, with a few notable differences, the structure of *Dm*-CI is like that of mammals, validating its use as a

genetically tractable model for the study of metazoan CI physiology. Given that inhibitors of CI have been developed as potential agricultural pesticides (*Murai and Miyoshi, 2016*), this structure will be a valuable resource for the development of more selective inhibitors. Due to its close relatedness to *D. melanogaster* (*Table 2*), our structure is of particular value to develop targeted pesticides against spotted-wing drosophila (*D. suzukii*), a major invasive agricultural pest of the berry and wine industry in Southeast Asia, Europe, and America (*Tait et al., 2021*). More broadly, the unique features of *Dm*-CI revealed here suggest strategies for the development of insecticides that could help control insect vectors of human disease. This study highlights the utility of diverse model organisms in the study of important biochemical processes as we learn as much from the differences as we do from the similarities. In addition, our structures reveal unanticipated mechanisms that have evolved to regulate the assembly and activity of mitochondrial CI that may be exploited to modulate assembly or activity in other organisms. Additional studies on *Dm*-CI as well as other species are needed to fully understand the different mechanisms which have evolved to regulate the assembly and activity of this important enzyme.

## Materials and methods

### *Drosophila* stocks and husbandry

*Drosophila* strains were maintained in vials containing agar, yeast, molasses, and cornmeal medium supplemented with propionic acid and methylparaben in humidified environmental chambers (Forma environmental chambers) on a 12 hr:12 hr dark: light cycle. Mitochondrial preparations used for structure determination were from female $w^{1118}$ flies. To examine the extent of incorporation of NDUFA2-FLAG into CI, genetic crosses were set up between female flies of the genotype, *y w; Mhc-Gal4* and *UAS-NDUFA2-FLAG* males at 25 °C. After the *Mhc-Gal4/UAS-NDUFA2-FLAG* flies eclosed, they were maintained at 25 °C for two days, prior to dissection of thoraces. *Mhc-Gal4 /w^{1118}* flies were used as controls.

### Mitochondria purification

Mitochondrial purification was performed as previously described (*Rera et al., 2011*). Briefly, approximately 2400 fly thoraces were dissected and gently crushed with a Dounce homogenizer in 1 mL per 20 thoraces of pre-chilled mitochondrial isolation buffer containing 20 mM HEPES-KOH pH 7.5, 0.6 M Sorbitol, 1 mM EDTA, 1 mM DTT, 0.1 mg/ml BSA, 10 units/ml Trasylol, and 0.5 mM PMSF on ice. After two rounds of centrifugation at 500 × g for 5 min at 4 °C to remove insoluble material, the supernatant was recovered and centrifuged at 5000 × g for 20 min at 4 °C. The pellet which is enriched for mitochondria was washed twice in the mitochondrial isolation buffer and stored at –80 °C until further processing.

### Spectroscopic activity assays

Activity from *D. melanogaster* or porcine mitochondrial membranes were measured in reaction buffer (20 mM HEPES, pH 7.4, 50 mM NaCl, 10% glycerol (v/v), 0.1% BSA (w/v), 0.1% CHAPS (w/v), 0.1% LMNG (w/v), 100 µM DQ) at 27 ± 1.30 µg/ml or 20 ± 0.21 µg/ml, respectively by spectroscopic observation of the oxidation of NADH (200 µM) at 340 nm in 1 ml cuvettes at room temperature using a Molecular Devices (San Jose, CA) Spectramax M2 spectrophotometer. The membrane samples were mixed with reaction buffer by tumbling and aliquoted into the 1 ml cuvette to a final volume of 1 ml. Activity of detergent-solubilized partially purified samples (see below for purification methodology) of *D. melanogaster* CI (1.34 ± 0.03 µg/ml) or *S. scrofa* SC I+III$_2$ (2.75 ± 0.15 µg/ml) was measured as above. Measurements of the initial rates were done in 3–5 replicates, averaged and background corrected.

For the MgCl$_2$ assay, the tubes were incubated at 37 °C for 30 min, after which 5 µM NADH or an equivalent amount of buffer was added to the tube and mixed by pipetting. 30 s after this addition, 5 mM MgCl$_2$ or water was added to the corresponding tubes and mixed by pipetting. The reaction mix in the tubes was transferred to cuvettes and the reactions were started by the addition of 200 µM NADH and briefly mixed by pipetting before recording every 2 s for 10 min (cuvettes).

For the NEM assay, the plate/tubes was/were incubated at 37 °C for 20/30 min, after which 5 µM NADH or an equivalent amount of buffer was added to the well/tube and mixed by pipetting. 30 s

**Table 2.** Sequence homology between CI subunits of *D. melanogaster* and *D. suzukii*.

| Subunit Name | Uniprot Annotation | *Ds* Homology with *Dm* (%) |
|---|---|---|
| NDUFV1 | NADH dehydrogenase [ubiquinone] flavoprotein 1, mitochondrial | 94.30% |
| NDUFV2 | NADH dehydrogenase (Ubiquinone) 24 kDa subunit, isoform A | 97.93% |
| NDUFS1 | NADH-ubiquinone oxidoreductase 75 kDa subunit, mitochondrial | 96.85% |
| NDUFS2 | Complex I-49kD | 91.03% |
| NDUFS3 | NADH dehydrogenase [ubiquinone] iron-sulfur protein 3, mitochondrial | 97.58% |
| NDUFS7 | LD31474p/NADH dehydrogenase (Ubiquinone) 20 kDa subunit, isoform A | 96.85% |
| NDUFS8 | NADH dehydrogenase (ubiquinone) 23 kDa subunit | 96.77% |
| ND1 | NADH-ubiquinone oxidoreductase chain 1 | 98.13% |
| ND2 | NADH-ubiquinone oxidoreductase chain 2 | 97.06% |
| ND3 | NADH-ubiquinone oxidoreductase chain 3 | 99.00% |
| ND4 | NADH-ubiquinone oxidoreductase chain 4 | 95.57% |
| ND4L | NADH-ubiquinone oxidoreductase chain 4 L | 98.96% |
| ND5 | NADH-ubiquinone oxidoreductase chain 5 | 94.23% |
| ND6 | NADH-ubiquinone oxidoreductase chain 6 | 93.10% |
| NDUFA1 | Complex I-MWFE | 98.61% |
| NDUFA2 | NADH dehydrogenase (Ubiquinone) B8 subunit | 91.58% |
| NDUFA3 | uncharacterized protein Dmel_CG9034, isoform B | 97.40% |
| NDUFA5 | NADH dehydrogenase (Ubiquinone) 13 kDa B subunit | 93.55% |
| NDUFA6 | Complex I-B14 | 100.00% |
| NDUFA7 | Complex I-B14.5a | 91.35% |
| NDUFA8 | NADH dehydrogenase [ubiquinone] 1 alpha subcomplex subunit 8 | 96.57% |
| NDUFA9 | NADH dehydrogenase (Ubiquinone) 39 kDa subunit, isoform A | 99.04% |
| NDUFA10 | NADH dehydrogenase [ubiquinone] 1 alpha subcomplex subunit 10 | 92.87% |
| NDUFA11 | Complex I-B14.7 | 95.81% |
| NDUFA12 | NADH dehydrogenase [ubiquinone] 1 alpha subcomplex subunit 12 | 96.48% |
| NDUFA13 | NADH dehydrogenase [ubiquinone] 1 alpha subcomplex subunit 13 | 98.05% |
| NDUFAB1-α | Acyl carrier protein | 96.71% |
| NDUFAB1-β | Acyl carrier protein | 96.71% |
| NDUFB1 | Complex I-MNLL | 91.07% |
| NDUFB2 | GEO11417p1/NADH dehydrogenase (Ubiquinone) AGGG subunit, isoform A | 72.83% |
| NDUFB3 | Complex I-B12 | 90.70% |
| NDUFB4 | Complex I-B15 | 93.81% |
| NDUFB5 | Complex I-SGDH | 93.55% |
| NDUFB6 | Complex I-B17 | 94.55% |
| NDUFB7 | NADH dehydrogenase [ubiquinone] 1 beta subcomplex subunit 7 | 94.02% |
| NDUFB8 | NADH dehydrogenase [ubiquinone] 1 beta subcomplex subunit 8 | 92.00% |
| NDUFB9 | NADH dehydrogenase [ubiquinone] 1 beta subcomplex subunit 9 | 90.97% |
| NDUFB10 | NADH dehydrogenase [ubiquinone] 1 beta subcomplex subunit 10 | 96.86% |

*Table 2 continued on next page*

*Table 2 continued*

| Subunit Name | Uniprot Annotation | *Ds* Homology with *Dm* (%) |
|---|---|---|
| NDUFB11 | Complex I-ESSS | 90.67% |
| NDUFC2 | NADH dehydrogenase [ubiquinone] 1 subunit C2 | 87.07% |
| NDUFS4 | NADH dehydrogenase [ubiquinone] iron-sulfur protein 4 | 87.10% |
| NDUFS5 | Complex I-15 kDa | 100.00% |
| NDUFS6 | NADH dehydrogenase [ubiquinone] iron-sulfur protein 6 | 91.27% |
| NDUFV3 | NADH dehydrogenase [ubiquinone] flavoprotein 3, mitochondrial | 72.48% |

after this addition, 0.5 mM NEM, 2 mM NEM or water was added to the corresponding wells/tubes and mixed by pipetting after which the plate/tubes was/were incubated for 20 min in dark at room temperature. The reaction mix in the tubes was transferred to cuvettes and the reactions in the cuvettes/plates were started by the addition of 200 µM NADH and briefly mixed by pipetting before recording every 2 s for 5 min (plates) /10 min (cuvettes).

## Western blotting

Western blotting was performed as previously described (*Murari et al., 2022*). Briefly, following the separation of protein complexes on 3–12% precast Bis-Tris Native PAGE gels (Life Technologies), the proteins were transferred to polyvinylidene difluoride (PVDF) membranes (Bio-Rad). Subsequently, the PVDF membrane was blocked in 5% (wt/vol) nonfat dry milk (NFDM) in tris-buffered saline (TBS) for 30 min and incubated in the appropriate primary antibody dissolved in 2% BSA and 0.1% Tween 20 in TBS (TBST) overnight at 4 °C. Subsequently, the blot was rinsed four times for 10 min each in 0.1% TBST, blocked for 30 min in 5% (wt/vol) NFDM in TBST, and incubated for 2 hr at room temperature with the appropriate HRP-conjugated secondary antibody dissolved in 2% BSA and 0.1% TBST. Afterward, samples were rinsed four times for 10 min each in 0.1% TBST. Immunoreactivity was detected by a SuperSignal West Pico PLUS Chemiluminescent kit (Thermo Scientific, 34578) and analyzed by a ChemiDoc gel imaging system from Bio-Rad. The primary antibodies used were anti-NDUFS3 (Abcam, ab14711), anti-NDUFA2 (this study), anti-FLAG (MilliporeSigma, F3165), and anti-ATPsynß (Life Technologies, A21351). Secondary antibodies used were goat anti-rabbit horseradish peroxidase (PI31460 from Pierce) and goat anti-mouse horseradish peroxidase (PI31430 from Pierce). To generate a rabbit polyclonal antibody for the *Drosophila* ortholog of NDUFA2 (CG15434) the following synthetic peptide was used: DPKGDTSKGVREYVER-Cys.

## Electron transport chain complex purification

The following operations were carried out at 4 °C unless otherwise indicated. The mitochondria pellet was resuspended and lysed in milli-Q water at 10 mL/g (of starting mitochondria, wet weight) using a Dounce homogenizer, to which KCl was added to a final concentration of 150 mM. The mitochondrial membrane was pelleted by centrifugation at 32,000 × g for 45 min and washed once in buffer M10 (20 mM Tris pH 7.4, 50 mM NaCl, 1 mM EDTA, 2 mM dithiothreitol (DTT), 0.002% PMSF (w/v), 10% glycerol (v/v)) at 18 mL/g (of starting mitochondria). The resulting membrane pellet was resuspended in buffer M10 at 3 mL/g (of starting mitochondria) and the protein concentration was determined using a BCA assay (Pierce Thermo Fisher). The resuspended membranes were stored at 10 mg/mL of total protein at –80 °C in a final glycerol concentration of 30% (v/v) after dilution with buffer M90 (20 mM Tris pH 7.4, 50 mM NaCl, 1 mM EDTA, 2 mM DTT, 0.002% PMSF (w/v), 90% glycerol (v/v)). Usual yield was ~30 mg total membrane protein per gram of *D. melanogaster* thorax.

The thawed mitochondrial membrane resuspension was solubilized in buffer MX (30 mM HEPES pH 7.7, 150 mM potassium acetate, 0.002% PMSF, 10% (v/v) glycerol) by slow tumbling for 1 hr at 4 °C with 1% digitonin (w/v) at a detergent-to-protein ratio of 4:1 (w/w). The insoluble material was cleared by centrifugation at 16,000 × g for 20 min. Amphipol A8-35 was added to the supernatant to a final concentration of 0.3% (w/v), before incubation with slow tumbling for 1 hr at 4 °C. Digitonin was removed from the supernatant by dialyzing the sample first in a buffer containing γ-cyclodextrin

followed by dialysis in buffer containing Bio-beads. The dialyzed sample was centrifuged at 16,000 × g for 20 min to remove any precipitate. The supernatant was concentrated in 100 kDa MWCO centrifugal concentrators to 0.250 mL and loaded onto a continuous 15 to 45% (w/v) sucrose gradient in SGB buffer (15 mM HEPES pH 7.8, 20 mM KCl). After centrifugation in an SW40Ti swinging-bucket rotor at 149,176 × g for 24 hr, the sucrose gradients were fractionated using a Biocomp gradient profiler. Fractions were assayed for CI activity by running them on a 3–12% Tris-glycine blue-native PAGE (BN-PAGE) gel and a nitrotetrazoleum blue in-gel assay was performed as previously described (*Maldonado et al., 2020*). Fractions displaying CI activity were pooled and concentrated to a final concentration of 5 mg/mL.

Partial purification of the complexes for activity assays was carried out as above with slight modifications. Briefly, following digitonin extraction, the insoluble material was cleared by centrifugation at 16,000 × g for 20 min. The supernatant was concentrated in 100 kDa MWCO centrifugal concentrators to 0.250 ml and loaded onto a continuous 20 to 45% (w/v) sucrose gradient in SGB buffer with 0.01% GDN (15 mM HEPES pH 7.8, 20 mM KCl, 0.01% GDN). After centrifugation in an SW40Ti swinging-bucket rotor at 149,176 × g for 24 hr, the sucrose gradients were fractionated using a Biocomp gradient profiler. Fractions were assayed for CI activity by running them on a 3–12% Tris-glycine blue-native PAGE (BN-PAGE) gel and a nitrotetrazoleum blue in-gel assay was performed as previously described (*Maldonado et al., 2020*). Fractions displaying CI activity were pooled, and buffer was exchanged into buffer containing 0.005% GDN (30 mM HEPES pH 7.8, 150 mM Potassium acetate, 0.005% GDN) and concentrated to a final concentration of 3–5 mg/ml.

## CryoEM grid preparation and data collection

Four microliters of concentrated fractions from the sucrose gradients were applied onto a Quantifoil R1.2/1.3 300 mesh copper grid glow-discharged at 30 mA for 30 s before sample application. In a GP2, the grid was first incubated for 20 s at 100% humidity, then blotted for 4 s before plunge-freezing into liquid ethane cooled by liquid nitrogen. A total of 11,065 movies were collected using SerialEM on a 200 kV ThermoFisher Glacios microscope equipped with a Gatan Quantum K3 detector, at a nominal magnification of 56,818 (0.44 Å/pixel under super-resolution mode). A dose of 20 electrons/Å$^2$/s with a 3 s exposure time was fractionated into 75 frames for each movie.

## CryoEM image processing

The raw movies were binned twofold and motion-corrected using the MotionCor2 (*Zheng et al., 2017*), followed by per-micrograph contrast transfer function (ctf) estimation using the CTFFIND4.1 (*Rohou and Grigorieff, 2015*), both implemented in Relion 3.1.0 (*Zivanov et al., 2018a*). Micrographs were then curated to remove images lacking high-resolution ctf correlations. Particles were picked using crYOLO (*Wagner et al., 2019*). The initial 698,452 picked particles were extracted in Relion 3.1.0 with 512 pixel$^2$ boxes, followed by 2D classification, 3D *ab initio* reconstruction, and 3D refinement in cryoSPARC v3.2.0 (*Punjani et al., 2017*). Iterative 2D classification and 3D *ab initio* reconstruction resulted in 293,389 good particles corresponding to Dm-CI, 25,080 particles corresponding to Dm-CIII, and 31,198 particles corresponding to Dm-CV (*Figure 1—figure supplement 2*). Homogenous refinement followed by non-uniform refinement (*Punjani et al., 2020*) of *Dm*-CI in cryoSPARC resulted in an initial reference map of 3.71 Å. This particle set was then transferred back into Relion 3.1.0 for further processing involving several rounds of global search, CTF refinement, Bayesian polishing (*Zivanov et al., 2018b*) and local searches resulting in a final map of 3.44 Å. This map was used for the initial model building in Coot (*Emsley et al., 2010*). Following initial model building and refinement in Phenix (*Liebschner et al., 2019*), masks corresponding to the peripheral arm, membrane arm, and the whole CI were generated in Relion. Iterative masked refinement and 3D classification resulted in a final reference map of 3.30 Å of the *Dm*-CI after import back into cryoSPARC for non-uniform refinement.

Poor local resolution and broken density around at the matrix interface of the MA and PA (NDUFA10, NDUFA5, NDUFA6, NDUFS4, NDUFAB1-α) of CI prompted us to further classify CI particles using a mask around the hinge region. 3D classification of CI particles using a mask around the hinge region resulted in two distinct classes of CI particles. Iterative homogenous refinement and non-uniform refinement of the classes resulted in reference maps of 3.40 Å for both classes. The final focused map was post-processed using DeepEMhancer (*Sanchez-Garcia et al., 2021*) which improved the

connectivity of certain regions of protein but also removed density for structured lipids. All software suites used for data processing and refinement except for cryoSPARC were accessed through the SBGrid consortium (*Morin et al., 2013*). 3D variability analysis (3DVA) on all 239,389 good particles was performed in cryoSPARC (*Punjani and Fleet, 2020*) to solve for three eigen volumes of the 3D covariance. Volume series corresponding to each of the components is generated in cryoSPARC. Molecular graphics and analyses were performed with UCSF ChimeraX, developed by the Resource for Biocomputing, Visualization, and Informatics at the University of California, San Francisco, with support from National Institutes of Health R01-GM129325 and the Office of Cyber Infrastructure and Computational Biology, National Institute of Allergy and Infectious Diseases (*Pettersen et al., 2021*).

## Model building and refinement

All manual model building was performed in Coot 0.9.2 (*Emsley et al., 2010*) and refinements were performed in Phenix-1.19.1 (*Liebschner et al., 2019*). Mammalian CI was docked into the *Dm*-CI map and Alpha-fold models (*Jumper et al., 2021*), accessed via Uniprot (*Bateman et al., 2020*), of *Dm*-CI subunits, were structurally aligned to each of the corresponding mammalian CI subunits to generate an initial model of the *Dm*-CI. The model-map fit was manually inspected, and the model was rebuilt where necessary to generate an initial *Dm*-CI model. Secondary structure restraints were first automatically generated from the manually built model, then edited according to the outcome of the Phenix refinement. Bond length and angle restraints for metal ion coordination and amino acid side chain linkage were generated manually, and a ligand.cif file was also provided for non-default ligands in Phenix. The refined model was manually inspected and edited in Coot before the next round of Phenix refinement, and this iterative cycle continued until the model statistics converged before the submission of maps and models to the EMDB and PDB databases. Model statistics and details by subunit are provided in *Table 1*.

## Acknowledgements

The data were collected at the UC Davis BioEM Core facility. We thank Dr. Fei Guo for assistance with data collection. We thank Dr. María Maldonado for critical reading of the manuscript. This work was funded by the NIGMS of the NIH under Awards R35GM137929 (JAL), and R35GM124717 (EO-A), and by the NIAMS of the NIH under Award R21AR077312 (EO-A). The content is solely the responsibility of the authors and does not necessarily represent the official views of the National Institutes of Health.

## Additional information

### Funding

| Funder | Grant reference number | Author |
|---|---|---|
| National Institute of General Medical Sciences | R35GM137929 | James A Letts |
| National Institute of General Medical Sciences | R35GM124717 | Edward Owusu-Ansah |
| National Institute of Arthritis and Musculoskeletal and Skin Diseases | R21AR077312 | Edward Owusu-Ansah |

The funders had no role in study design, data collection and interpretation, or the decision to submit the work for publication.

### Author contributions

Abhilash Padavannil, Formal analysis, Validation, Investigation, Visualization, Methodology, Writing – original draft, Writing – review and editing; Anjaneyulu Murari, Shauna-Kay Rhooms, Methodology; Edward Owusu-Ansah, Conceptualization, Supervision, Funding acquisition, Validation, Visualization, Methodology, Writing – original draft, Project administration, Writing – review and editing; James A

Letts, Conceptualization, Data curation, Formal analysis, Supervision, Funding acquisition, Validation, Visualization, Writing – original draft, Project administration, Writing – review and editing

### Author ORCIDs
Abhilash Padavannil  http://orcid.org/0000-0002-9949-6776
Anjaneyulu Murari  http://orcid.org/0000-0002-7532-964X
Edward Owusu-Ansah  http://orcid.org/0000-0002-3451-1752
James A Letts  http://orcid.org/0000-0002-9864-3586

### Decision letter and Author response
Decision letter https://doi.org/10.7554/eLife.84415.sa1
Author response https://doi.org/10.7554/eLife.84415.sa2

## Additional files

### Supplementary files
• MDAR checklist

### Data availability
Single-particle cryogenic electron micrograph movies are available on the Electron Microscopy Public Image Archive, accession code EMPIAR-11272. The maps and models are available on the Electron Microscopy Database (EMDB) and Protein Data Bank (PDB). The accession codes for the Helix-locked state are EMDB-28582, PDB-8ESZ and for the flexible class 1 state are EMDB-28581, PDB-8ESW.

The following datasets were generated:

| Author(s) | Year | Dataset title | Dataset URL | Database and Identifier |
| --- | --- | --- | --- | --- |
| Letts JA, Padavannil A | 2022 | Mitochondrial complex I from *Drosophila melanogaster* | https://www.ebi.ac.uk/empiar/ | EMPIAR, 11272 |
| Letts JA, Padavannil A | 2022 | Structure of mitochondrial complex I from *Drosophila melanogaster*, Helix-locked state | https://www.ebi.ac.uk/emdb/search/EMDB-28582 | Electron Microscopy Data Bank, EMD-28582 |
| Letts JA, Padavannil A | 2022 | Structure of mitochondrial complex I from *Drosophila melanogaster*, Flexible class 1 | https://www.ebi.ac.uk/emdb/search/EMDB-28581 | Electron Microscopy Data Bank, EMD-28581 |
| Letts JA, Padavannil A | 2022 | Structure of mitochondrial complex I from *Drosophila melanogaster*, Helix-locked state | https://www.rcsb.org/structure/unreleased/8ESZ | RCSB Protein Data Bank, 8ESZ |
| Letts JA, Padavannil A | 2022 | Structure of mitochondrial complex I from *Drosophila melanogaster*, Flexible class 1 | https://www.rcsb.org/structure/unreleased/8ESW | RCSB Protein Data Bank, 8ESW |

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
