## [Editor Report]

This important study offers new insights into the structure and function of respiratory complex I. Based on convincing cryoEM data for the enzyme complex from the insect model organism *Drosophila melanogaster*, the authors discuss the functional significance of two major conformational states. This study is relevant to readers interested in the evolution and molecular mechanisms of respiratory chain complexes, as well as mitochondrial diseases and the development of new insecticides.

---

## [Decision Letter]

**Decision letter after peer review:**

Thank you for submitting your article "Resting mitochondrial complex I from *Drosophila melanogaster* adopts a helix-locked state." for consideration by *eLife*. Your article has been reviewed by 3 peer reviewers, and the evaluation has been overseen by a Reviewing Editor and Volker Dötsch as the Senior Editor. The following individual involved in the review of your submission has agreed to reveal their identity: Janet Vonck (Reviewer #1).

Essential revisions:

1. The evolutionary aspect should be clarified: Opisthokonts include Protostomia and Deuterostomia but yeasts (with the A/D present) are not Protostomia.

2. Please report the Q reductase activity of purified complex I in µmol min-1 mg-1 and electrons s^-1^ to allow straightforward comparison with other enzyme preparations. Please add assay conditions for the purified enzyme to the Materials and methods section.

3. The absence of an A/D transition in Dm complex I is based on the insensitivity of the Q reductase activity to NEM. The NEM assay requires that a specific cysteine in ND3 is exclusively accessible in the D state and that this state is arrested after NEM modification. It seems possible that Dm complex I undergoes an A/D transition while the reaction of NEM with the ND3 cysteine is slow or impossible throughout. A new type of deactive state is even considered by the authors and described as a helix-locked-open state. Please show a trace of a typical measurement of activity to allow evaluation of whether or not a lag phase is present. More conditions for the reaction of NEM with complex I should be tested. Please also evaluate the impact of divalent cations on the activity as an alternative assay for the A/D transition (Biochimica et Biophysica Acta, 1098 (1992) 144-150).

4. Interpretation of the different conformations is admittedly difficult at this stage. The classification of the locked-open state as an off-pathway species needs more explanation.

5. Terminology regarding conformational states should be clearly defined and used consistently, i.e. active/deactive, locked/open, tensed/relaxed.

*Reviewer #1 (Recommendations for the authors):*

The main concern is the interpretation of the "lock helix" on p. 18: "Therefore, it is more likely that the closed state in which this loop is freer to undergo conformational changes is an on-pathway state and that the helix-locked open state is an off-pathway resting state, similar in function but biochemically distinct from the D state of yeast and mammals." This may sound reasonable, but the D (deactive) state of yeast and mammals have in fact disordered ND3 loops. This appears to be the only argument for the assignment of the locked-open state as deactive, so it either needs experimental evidence or other possibilities should be considered.

p. 13 lines 4-5. The nomenclature "closed" and "locked open" is confusing, as it turns out that the states are more similar to the mammalian states of the opposite name. Also "locked open" sounds a bit strange, more intuitive might be "wedged open", but this still includes "open". Open and closed are anyway not great terms to describe a very small change in the angle between the arms. Solutions may be to use a neutral term (like A and B) or relate to the order/disorder of the helix.

Related to the above points, the title of the paper should be reconsidered.

P. 2 line 7: "In metazoans, the generation of the pmf is driven by the transfer of electrons…" This is not unique to metazoans. Better to leave the first two words out.

p. 3 line 3 "a wave of conformational changes". Actually, proton transfer in the MA appears not to involve conformational changes. Please rephrase.

Page 6: Dm complex I has 29 accessory subunits, 2 fewer than mammals. But it's not clear which two they are. One of them is NDUFC1. NDUFA2 is missing from the structure, but this is assumed to have been lost during purification. Is this one meant, or is another subunit truly absent? It may be more logical to first discuss the two missing subunits and then the sub-stoichiometric NDUFV3 and the identification of the NDUFA3 ortholog.

p. 9 line 31 typo: know should be known.

p. 10 line 7. "NDUFB5, NDUFB8 and NDUFS5…" For the benefit of the reader who does not know all subunit locations, it would be useful to write: "the accessory subunits NDUFB5, NDUFB8, and NDUFS5 on the IMS side…"

P. 12 line 5 typo: Lipids from… should be Lipids form…

p. 13 line 25 typo: close state should be closed state.

p. 15 line 19 typo: structure should be structure of.

Figure 1 – supplement 2: the last of the 8 class averages appears not the be complex V (ATP synthase) but vacuolar ATPase judging by its larger head, multiple peripheral stalks, and broader membrane domain. It may be more accurate to label this panel "ATPases" instead of "CV".

Figure 2A: it would be useful to indicate the residues flanking the disordered loop in *Drosophila* NDUFS1 to give an indication of how long this stretch is.

*Reviewer #3 (Recommendations for the authors):*

General comments:

1. The following statements are not consistent as well as the beginning of the Discussion section.

Abstract: "Biochemical studies have found a divide in the behavior of complex I in metazoans that aligns with the evolutionary split between Protostomia and Deuterostomia. Complex I from Deuterostomia including mammals can adopt an off-pathway "deactive" state, whereas complex I from Protostomia cannot."

Page 3. Lines 9-10. "Complicating the structural elucidation of the coupling mechanism is the fact that resting CI from opisthokonts (yeast and mammals) have been shown to exist in two distinct biochemical states…"

While this is a tempting suggestion, it contradicts the facts that Protostomia (no A/D transition) are Opisthokonts and that the A/D transition has been found in yeasts (Yarrowia or Neurospora) which are Opisthokonts and neither Protostomia nor Deuterostomia. Therefore, phylogenetics does not fully explain why the A/D is present in the Deuterostomia _and_ the yeast. The quoted statements and the beginning of the Discussion might benefit from some rephrasing.

2. Page 20 Lines 5-7. "…the structure of Dm-CI provides the first structure of a protostomian CI that does not share the standard A-to-D transition as defined biochemically…"

Abstract: "…complex I does not deactivate…"

This is not clear. Is it because of the lesser sensitivity to NEM or because of the differences observed by CryoEM? What is the definition of the D state? This reviewer has several points the authors could clarify in their manuscript: is thiol residues of the TMH1-2 ND3 loop assessable for NEM or not? If it can be modified why it does not inhibit the activity of the D.m enzyme and inhibits the mammalian one? If D.m. complex I adopts a more "open-like state" or "resting" form (i.e. resembling the D-form) why it is not sensitive to NEM treatment? This should be openly discussed to increase the significance and visibility of this publication.

3. The main statement of the paper that Protostomia complex I does not display any A/D transitions is based on the data shown in Figure 1C in the presented manuscript. The publication by Maklashina 2003 where the absence of the A/D transition in crickets was shown is another single piece of evidence supporting that hypothesis. While a significant body of work on high-quality CryoEM analysis has been performed, kinetic characterization of the NEM sensitivity is somehow very limited. Maybe insect enzyme requires a longer time for deactivation/activation? While this reviewer agrees that insect complex I does not display the A/D transition as originally defined for the bovine enzyme, a better characterization of the NEM effect is desirable with more conditions tested (temperature, incubation time, NADH treatment, etc). For the kinetic assay, does the presence of 0.1%CHAPS and 0.1% digitonin have no effect on the NADH:DQ reductase reaction of complex I? Also, the difference between the D and the A forms after NEM treatment most likely would be more pronounced if the concentration of DQ is decreased to 50-70 microM.

---

## [Author Response]

Essential revisions:1. The evolutionary aspect should be clarified: Opisthokonts include Protostomia and Deuterostomia but yeasts (with the A/D present) are not Protostomia.

This part of the introduction has been rewritten for clarity and to address the specific comments of the reviewers below.

Please see – Page 4 lines 1-12.

2. Please report the Q reductase activity of purified complex I in µmol min-1 mg-1 and electrons s^-1^ to allow straightforward comparison with other enzyme preparations. Please add assay conditions for the purified enzyme to the Materials and methods section.

Although we did not fully purify the sample as indicated by the presence of other respiratory components on our grids, we have included the NADH oxidation activity of our Dm-CI samples in µmol min-1 mg-1 as well as for a porcine complex I sample prepared according to the same protocol to allow for direct comparison. The goal of the study was not to purify Dm-CI to homogeneity but to characterize the complex functionally and structurally. As the complex was not purified to homogeneity the activity in electrons s^-1^ cannot be determined.

Please see – Page 5 line 31 and Page 6 lines 1-2 and “Spectroscopic activity assays” subsection in the Materials and methods section.

3. The absence of an A/D transition in Dm complex I is based on the insensitivity of the Q reductase activity to NEM. The NEM assay requires that a specific cysteine in ND3 is exclusively accessible in the D state and that this state is arrested after NEM modification. It seems possible that Dm complex I undergoes an A/D transition while the reaction of NEM with the ND3 cysteine is slow or impossible throughout. A new type of deactive state is even considered by the authors and described as a helix-locked-open state. Please show a trace of a typical measurement of activity to allow evaluation of whether or not a lag phase is present. More conditions for the reaction of NEM with complex I should be tested. Please also evaluate the impact of divalent cations on the activity as an alternative assay for the A/D transition (Biochimica et Biophysica Acta, 1098 (1992) 144-150).

We thank the reviewers for this excellent suggestion and have performed the requested experiments and present them in the Figure 1 D-G and Figure 1 figure supplement 1C-F. These data indeed indicate that after incubation at elevated temperatures in the absence of substrate, Dm-CI does have slow activation kinetics. However, in contrast to porcine CI the activation kinetics are not sensitive to NEM or MgCl2. These new results are presented in the A-to-D transition subsection which we have renamed “*D. melanogaster* CI possesses an NEM insensitive off-pathway resting state.”

Please see – Results section “*D. melanogaster* CI possesses an NEM insensitive off-pathway resting state.”

4. Interpretation of the different conformations is admittedly difficult at this stage. The classification of the locked-open state as an off-pathway species needs more explanation.

We agree with the reviewers and have changed the discussion to focus less on one possible interpretation but instead discuss all major possibilities and suggest future experiments.

Please see – Discussion section.

5. Terminology regarding conformational states should be clearly defined and used consistently, i.e. active/deactive, locked/open, tensed/relaxed.

We agree with the reviewers though this may be a comment for the field more generally as opposed to one manuscript. Nonetheless, we have re-written portions of the results and discussion to clarify our interpretation and comparison to other known states. We have also renamed our states to make the discussion and comparisons clearer.

Please see – Page 13 lines 29-30.

Reviewer #1 (Recommendations for the authors):The main concern is the interpretation of the "lock helix" on p. 18: "Therefore, it is more likely that the closed state in which this loop is freer to undergo conformational changes is an on-pathway state and that the helix-locked open state is an off-pathway resting state, similar in function but biochemically distinct from the D state of yeast and mammals." This may sound reasonable, but the D (deactive) state of yeast and mammals have in fact disordered ND3 loops. This appears to be the only argument for the assignment of the locked-open state as deactive, so it either needs experimental evidence or other possibilities should be considered.

We agree with the reviewer that we were too focused on one possible interpretation of our results. We have clarified the naming of the states and expanded the discussion to more evenly consider other possibilities.

Please see the updated Discussion section.

p. 13 lines 4-5. The nomenclature "closed" and "locked open" is confusing, as it turns out that the states are more similar to the mammalian states of the opposite name. Also "locked open" sounds a bit strange, more intuitive might be "wedged open", but this still includes "open". Open and closed are anyway not great terms to describe a very small change in the angle between the arms. Solutions may be to use a neutral term (like A and B) or relate to the order/disorder of the helix.Related to the above points, the title of the paper should be reconsidered.

We agree with the reviewer and have changed the names of the states to remove the terms “opened” and “closed” as these names have become unhelpful when used to compare complex I states by the relative positions of the peripheral and membrane arms and only seem to apply to the mammalian (or possibly other metazoan) enzymes.

We do however think the names “helix-locked” state and “flexible” state are appropriate (without reference to open or closed) as the presence of the helix locks the angle between the peripheral and membrane arms. We have clarified the description of this nomenclature in the manuscript.

Please see – Page 13 lines 29-30.

P. 2 line 7: "In metazoans, the generation of the pmf is driven by the transfer of electrons…" This is not unique to metazoans. Better to leave the first two words out.

Very true and done. Please see – Page 2 line 7.

p. 3 line 3 "a wave of conformational changes". Actually, proton transfer in the MA appears not to involve conformational changes. Please rephrase.

We have added “and electrostatic interactions” to better represent the proposed models, though we would argue that this is not settled. Please see – Page 3 lines 3-4.

Page 6: Dm complex I has 29 accessory subunits, 2 fewer than mammals. But it's not clear which two they are. One of them is NDUFC1. NDUFA2 is missing from the structure, but this is assumed to have been lost during purification. Is this one meant, or is another subunit truly absent? It may be more logical to first discuss the two missing subunits and then the sub-stoichiometric NDUFV3 and the identification of the NDUFA3 ortholog.

We have added the sentence “The two missing subunits are NDUFA2, which may have been lost during purification (see below) and NDUFC1.” Thanks!. Please see – Page 6 lines 26-27.

p. 9 line 31 typo: know should be known.

Corrected, thanks!

p. 10 line 7. "NDUFB5, NDUFB8 and NDUFS5…" For the benefit of the reader who does not know all subunit locations, it would be useful to write: "the accessory subunits NDUFB5, NDUFB8, and NDUFS5 on the IMS side…"

Added, thanks! Please see – Page 10 line 31.

P. 12 line 5 typo: Lipids from… should be Lipids form…

Corrected, thanks!

p. 13 line 25 typo: close state should be closed state.

Corrected, thanks!

p. 15 line 19 typo: structure should be structure of.

Corrected, thanks!

Figure 1 – supplement 2: the last of the 8 class averages appears not the be complex V (ATP synthase) but vacuolar ATPase judging by its larger head, multiple peripheral stalks, and broader membrane domain. It may be more accurate to label this panel "ATPases" instead of "CV".

Changed, thanks! Please see – Figure 1—figure supplement 2.

Figure 2A: it would be useful to indicate the residues flanking the disordered loop in *Drosophila* NDUFS1 to give an indication of how long this stretch is.

Added, thanks! Please see updated Figure 2A.

Reviewer #3 (Recommendations for the authors):General comments:1. The following statements are not consistent as well as the beginning of the Discussion section.Abstract: "Biochemical studies have found a divide in the behavior of complex I in metazoans that aligns with the evolutionary split between Protostomia and Deuterostomia. Complex I from Deuterostomia including mammals can adopt an off-pathway "deactive" state, whereas complex I from Protostomia cannot."Page 3. Lines 9-10. "Complicating the structural elucidation of the coupling mechanism is the fact that resting CI from opisthokonts (yeast and mammals) have been shown to exist in two distinct biochemical states…"While this is a tempting suggestion, it contradicts the facts that Protostomia (no A/D transition) are Opisthokonts and that the A/D transition has been found in yeasts (Yarrowia or Neurospora) which are Opisthokonts and neither Protostomia nor Deuterostomia. Therefore, phylogenetics does not fully explain why the A/D is present in the Deuterostomia _and_ the yeast. The quoted statements and the beginning of the Discussion might benefit from some rephrasing.

This discussion has been rephrased and clarified. Thanks!

Please see the updated Discussion section.

2. Page 20 Lines 5-7. "…the structure of Dm-CI provides the first structure of a protostomian CI that does not share the standard A-to-D transition as defined biochemically…"Abstract: "…complex I does not deactivate…"This is not clear. Is it because of the lesser sensitivity to NEM or because of the differences observed by CryoEM? What is the definition of the D state? This reviewer has several points the authors could clarify in their manuscript: is thiol residues of the TMH1-2 ND3 loop assessable for NEM or not? If it can be modified why it does not inhibit the activity of the D.m enzyme and inhibits the mammalian one? If D.m. complex I adopts a more "open-like state" or "resting" form (i.e. resembling the D-form) why it is not sensitive to NEM treatment? This should be openly discussed to increase the significance and visibility of this publication.

The discussion regarding the different states, comparison to other structures and the functional implications has been rephrased and clarified throughout. Thanks!

Please see the updated Discussion section.

3. The main statement of the paper that Protostomia complex I does not display any A/D transitions is based on the data shown in Figure 1C in the presented manuscript. The publication by Maklashina 2003 where the absence of the A/D transition in crickets was shown is another single piece of evidence supporting that hypothesis. While a significant body of work on high-quality CryoEM analysis has been performed, kinetic characterization of the NEM sensitivity is somehow very limited. Maybe insect enzyme requires a longer time for deactivation/activation? While this reviewer agrees that insect complex I does not display the A/D transition as originally defined for the bovine enzyme, a better characterization of the NEM effect is desirable with more conditions tested (temperature, incubation time, NADH treatment, etc). For the kinetic assay, does the presence of 0.1%CHAPS and 0.1% digitonin have no effect on the NADH:DQ reductase reaction of complex I? Also, the difference between the D and the A forms after NEM treatment most likely would be more pronounced if the concentration of DQ is decreased to 50-70 microM.

We thank the reviewer for this suggestion and have performed additional experiments. Indeed, we were able to identify an activation lag phase that is consistent with the presence of an off-pathway resting state. We have included this data in the Figure 1 D-G and Figure 1 figure supplement 1C-F and have updated the manuscript throughout to take into account these important new findings. We were not able to perform all of the suggested experiments due to limited *Drosophila* samples, but we have spent a great deal of time optimizing these assays using the mammalian enzymes and we can confidently state that the presence of the detergents do not negatively impact the electron transfer reaction of complex I. We agree that more NEM conditions should be tested, especially under turnover conditions, but we think that our additional experiments are sufficient to demonstrate the existence of an off-pathway resting state and further experiments though important for future work are beyond the scope of this manuscript.

Please see – Results section “*D. melanogaster* CI possesses an NEM insensitive off-pathway resting state.”